# Can Kernel Methods Explain How the Data Affects Neural Collapse?

**Vignesh Kothapalli**                                                                *vk2115@nyu.edu*
*Courant Institute of Mathematical Sciences*
*New York University*

**Tom Tirer**                                                                          *tirer.tom@gmail.com*
*Faculty of Engineering*
*Bar-Ilan University*

**Reviewed on OpenReview:** *https://openreview.net/forum?id=MbF1gYfIlY*

## Abstract

A vast amount of literature has recently focused on the "Neural Collapse" (NC) phenomenon, which emerges when training neural network (NN) classifiers beyond the zero training error point. The core component of NC is the decrease in the within-class variability of the network's deepest features, dubbed as NC1. The theoretical works that study NC are typically based on simplified unconstrained features models (UFMs) that mask any effect of the data on the extent of collapse. To address this limitation of UFMs, this paper explores the possibility of analyzing NC1 using kernels associated with shallow NNs. We begin by formulating an NC1 metric as a function of the kernel. Then, we specialize it to the NN Gaussian Process kernel (NNGP) and the Neural Tangent Kernel (NTK), associated with wide networks at initialization and during gradient-based training with a small learning rate, respectively. As a key result, we show that the NTK does not represent more collapsed features than the NNGP for Gaussian data of arbitrary dimensions. This showcases the limitations of data-independent kernels such as NTK in approximating the NC behavior of NNs. As an alternative to NTK, we then empirically explore a recently proposed *data-aware* Gaussian Process kernel, which generalizes NNGP to model feature learning. We show that this kernel yields lower NC1 than NNGP but may not follow the trends of the shallow NN. Our study demonstrates that adaptivity to data may allow kernel-based analysis of NC, though, further advancements in this area are still needed. A nice byproduct of our study is showing both theoretically and empirically that the choice of nonlinear activation function affects NC1 (with ERF yielding lower values than ReLU). The code is available at: https://github.com/kvignesh1420/shallow_nc1.

## 1 Introduction

Deep Neural Network classifiers are often trained beyond the zero training error point (Hoffer et al., 2017; Ma et al., 2018). In this regime, a phenomenon dubbed "Neural Collapse" (NC) emerges (Papyan et al., 2020). NC is typically described by the following components: (NC1) the networks' deepest features exhibit a significant decrease in the variability of within-class samples, (NC2) the mean features of different classes approach a certain symmetric structure, and (NC3) the last layer's weights become more aligned with the penultimate layer features' means. This behavior has been observed both when using the cross-entropy (CE) loss (Papyan et al., 2020) and the mean squared error (MSE) loss (Han et al., 2022).

Recently, a vast amount of literature has been dedicated to exploring NC (as surveyed in (Kothapalli, 2023)), studying the effect of imbalanced data (Fang et al., 2021; Thrampoulidis et al., 2022), depthwise evolution (Tirer & Bruna, 2022; Rangamani et al., 2023; Sukenik et al., 2023; He & Su, 2023), fine-grained structures

(Tirer et al., 2023; Yang et al., 2023; Kothapalli et al., 2023), and implications (Zhu et al., 2021; Galanti et al., 2022; Yang et al., 2022). Note that without a sufficient decrease in the features' within-class variability around the class means, measured by NC1 metrics, one may not gain valuable insights from the structure of the means of different classes. Therefore, oftentimes, NC papers focus specifically on NC1 rather than on other components of NC (Tirer et al., 2023; Yang et al., 2023; Kothapalli et al., 2023; Galanti et al., 2022; He & Su, 2023; Xu & Liu, 2023). Notably, most of the works that attempt to theoretically analyze the NC behavior (Mixon et al., 2020; Fang et al., 2021; Zhu et al., 2021; Han et al., 2022; Ji et al., 2022; Tirer & Bruna, 2022; Zhou et al., 2022; Thrampoulidis et al., 2022; Yaras et al., 2022; Tirer et al., 2023; Dang et al., 2023; Sukenik et al., 2023; Wojtowytsch et al., 2020) are based on variants of the unconstrained features model (UFM) (Mixon et al., 2020), which treats the deepest features of the training samples as free optimization variables. Therefore, such UFM analyses rely solely on the labels and cannot predict the effect of training data on the extent of collapse.

Since theoretically analyzing the behavior of (deep) NNs is challenging, simplifying approaches that are based on kernel methods (Schölkopf et al., 2002) have gained massive popularity (Neal, 1995; Lee et al., 2018; Jacot et al., 2018; Chizat et al., 2019; Arora et al., 2019; Lee et al., 2019). Prominent examples include the NN Gaussian Process kernel (NNGP) (Neal, 1995; Lee et al., 2018; Matthews et al., 2018), associated with the infinitely wide NNs at initialization, and the Neural Tangent Kernel (NTK) (Jacot et al., 2018), associated with their training in the "lazy regime", where the learning rate is sufficiently small. These approaches, and in particular NTK, were used to provide mathematical reasoning for deep learning phenomena such as achieving zero training loss (Jacot et al., 2018; Arora et al., 2019; Lee et al., 2019), faster learning of lower frequencies (Ronen et al., 2019), benefits of ResNets over fully connected networks (Huang et al., 2020; Tirer et al., 2022; Barzilai et al., 2022), usefulness of positional encoding in coordinated-based NNs (Tancik et al., 2020), and more. More recently, finite width variants of the aforementioned kernels have been studied (Hanin & Nica, 2019; Lee et al., 2020; Seroussi et al., 2023), aiming to mitigate the gap between practical NNs behavior and infinite width analyses (Woodworth et al., 2020; Ghorbani et al., 2020; Wei et al., 2019; Li et al., 2020).

**Why Kernels on Gaussian data for studying NC?**  In this paper, we provide a kernel-based analysis for NC1 (the core component of NC) to bypass the limitations of UFM-based analysis. More importantly, our work aims to highlight the role of data in the study of NC and thus leverages the Gaussian datasets for a systematic theoretical and empirical analysis of NC1 with kernels. This setup allows us to explore how different aspects of the data such as dimension, sample size, class imbalance, and linear separability affect the collapse in a controlled fashion. To this end, as also mentioned above, we focus on (1) *data-independent* kernels such as NNGP/NTK whose functional form does not change with data, and (2) *data-aware* GP kernels (Seroussi et al., 2023) which model the feature mappings learned by finite-width shallow NNs (especially shallow Fully Connected Networks (FCNs)). Our main contributions can be summarized as follows:

- Given an arbitrary kernel function, we establish expressions for the traces of the within- and between-class covariance matrices of the features — and consequently, an NC1 metric that depends on the features only through the kernel function.

- We specialize our kernel-based NC1 to kernels associated with shallow NNs. We analyze it for NNGP and NTK and show that, perhaps surprisingly, the NTK does not represent more collapsed features than the NNGP for Gaussian data.

- We analyze the *data-aware* GP kernel (Seroussi et al., 2023), which generalizes NNGP to model the feature mapping learned from the training data by shallow NNs. We present empirical results on Gaussian data to show that the adaptivity of such kernels yields lower NC1 and allows them to reasonably approximate the NC1 behavior of shallow NNs for linearly separable datasets. However, they may fail in regimes with imbalanced and non-linearly separable data in higher dimensions.

- A byproduct of our study is showing both theoretically and empirically that the choice of the nonlinear activation function affects NC1. Specifically, we show that using ERF yields lower NC1 (more collapsed features) than using ReLU.

**Paper Organization.** We review previous works on the UFM and kernels in the context of NC in Section 2. Section 3 introduces the setup based on 2 layer FCNs and kernels used throughout the paper. Section 4 introduces a generic NC1 metric for any arbitrary kernel function. This formulation is then used in Section 5 to theoretically and empirically analyze the NC1 of NNGP and NTK kernels. Note that the NTK feature map considers the FCN architecture and the GD optimization process (although not data-aware) for the NC1 analysis, and allows us to gain insights beyond NNGP (Seleznova et al., 2023). Owing to the limitations of the NTK, we wish to contrast NNGP with a kernel that takes into account both optimization and data. To this end, we transition to the recently introduced *data-aware* GP kernels by (Seroussi et al., 2023) for FCNs in Section 6. Finally, we discuss the limitations of such kernels in Section 7 and the conclusion in Section 8.

## 2   Related Work

The majority of the works that attempt to analyze NC behavior theoretically (Mixon et al., 2020; Fang et al., 2021; Zhu et al., 2021; Han et al., 2022; Ji et al., 2022; Tirer & Bruna, 2022; Zhou et al., 2022; Yaras et al., 2022; Dang et al., 2023; Sukenik et al., 2023) are based on variants of the UFM (Mixon et al., 2020). The work in (Tirer et al., 2023) attempts to generalize the UFM by adding a penalty term to the loss that ensures that the features matrix is in the vicinity of a predefined matrix. Yet, the model still lacks an explicit connection between the features and the data. In (Wang et al., 2023), the authors avoid optimizing the features directly but assume that the model is linear and that the data is nearly orthogonal, which are restrictive assumptions. In (Seleznova et al., 2023), the authors claim that having an exact class-wise block structure in the Gram matrix of the empirical NTK on training samples implies NC. Yet, they do not provide reasoning for reaching this collapsed Gram matrix and the analysis is still disconnected from the data.

Here, however, we fully depart from the UFM approach and provide analysis that explicitly depends on the data. Furthermore, unlike most works, our analysis applies to the less studied case where the data is class imbalanced (Fang et al., 2021; Yang et al., 2022; Dang et al., 2023; Thrampoulidis et al., 2022). Our kernel-based analysis utilizes results on NNGP and NTK from (Williams, 1996; Cho & Saul, 2009; Lee et al., 2018; Jacot et al., 2018; Lee et al., 2019). To simplify the analysis, we theoretically analyze kernels associated with shallow fully connected NNs. Focusing on shallow networks is justified by recent works demonstrating monotonic depthwise evolution of NC1 (Tirer & Bruna, 2022; Rangamani et al., 2023; Sukenik et al., 2023; He & Su, 2023; Tirer et al., 2023). Specifically, a data structure that promotes a larger reduction in NC1 for shallow NNs is expected to be more collapsed when using deep NNs. Since NC is related to training, and we show the limitation of NTK to capture it when compared to NNGP, we also utilize the generalization of NNGP that has been proposed in (Seroussi et al., 2023). In this *data-aware* GP kernel approach, there is an explicit kernel function that depends on the training data. Recently, this richer model has been used to study phase transition behaviors, such as grokking (Rubin et al., 2024), that cannot be captured with the data-independent kernels.

## 3   Problem Setup

• **Data:** We consider a dataset $\mathbf{X} \in \mathbb{R}^{d_0 \times N}$, comprising $N$ data points of dimension $d_0$ belonging to $C$ classes. Each class has size $n_c, c \in [C]$, where $[C] := \{1, 2, \cdots, C\}$ and $\sum_c n_c = N$. The dataset is represented in an "organized" matrix form as $\mathbf{X} = \begin{bmatrix} \mathbf{x}^{1,1} & \cdots & \mathbf{x}^{1,n_1}, \mathbf{x}^{2,1} \cdots & \mathbf{x}^{C,n_C} \end{bmatrix} \in \mathbb{R}^{d_0 \times N}$, where $\mathbf{x}^{c,i} \in \mathbb{R}^{d_0}$ represents the $i^{th}$ data point of the $c^{th}$ class. Specific assumptions on the data distribution will be presented during the paper together with the related theory or experiments.

• **Neural Network:** Unless stated otherwise, we consider a 2-layer fully connected neural network (2L-FCN) $\psi : \mathbb{R}^{d_0} \to \mathbb{R}^{d_2}$ with $l^{th}$ layer width $d_l, l \in \{1, 2\}$, and point-wise activation function $\phi(\cdot) : \mathbb{R} \to \mathbb{R}$. Let $\mathbf{W}^{(l)} \in \mathbb{R}^{d_l \times d_{l-1}}, \mathbf{b}^{(l)} \in \mathbb{R}^{d_l}$ denote the weight and bias parameters of the $l^{th}$ layer. At initialization, the entries $W_{ij}^{(l)}, b_i^{(l)}$ are drawn i.i.d from Gaussian distributions of mean 0 and variance $\sigma_w^2/d_{l-1}, \sigma_b^2$, respectively. For an input $\mathbf{x} \in \mathbb{R}^{d_0}$ to the network $\psi(\cdot)$, we denote the $i^{th}$ component of the output vector $\hat{y}_i(\mathbf{x}) \in \mathbb{R}$ as:

$$\hat{y}_i(\mathbf{x}) = b_i^{(2)} + \sum_{j=1}^{d_1} W_{ij}^{(2)} \phi\left(z_j(\mathbf{x})\right), \qquad z_j(\mathbf{x}) = b_j^{(1)} + \sum_{k=1}^{d_0} W_{jk}^{(1)} x_k. \tag{1}$$

• **Task:** We train the network $\psi(\cdot)$ to classify the data points $\mathbf{x}^{c,i}, c \in [C], i \in [n_c]$ to their respective classes. Let $\hat{\mathbf{Y}}, \mathbf{Y} \in \mathbb{R}^{d_2 \times N}$ denote the prediction and ground truth label matrices respectively:

$$\hat{\mathbf{Y}} = \begin{bmatrix} \hat{\mathbf{y}}^{1,1} & \cdots & \hat{\mathbf{y}}^{C,n_C} \end{bmatrix}, \qquad \mathbf{Y} = \begin{bmatrix} \mathbf{y}^{1,1} & \cdots & \mathbf{y}^{C,n_C} \end{bmatrix} \tag{2}$$

We aim to minimize the Mean Squared Error (MSE) between $\hat{\mathbf{Y}}, \mathbf{Y}$ using the following objective:

$$\mathcal{R}(\psi, \mathbf{X}, \mathbf{Y}) = \frac{1}{N} \left\| \hat{\mathbf{Y}} - \mathbf{Y} \right\|_F^2 + \lambda \sum_{l=1}^{2} \left( \left\| \mathbf{W}^{(l)} \right\|_F^2 + \left\| \mathbf{b}^{(l)} \right\|_F^2 \right). \tag{3}$$

Note that training classifiers with MSE loss has been shown to be a useful strategy (Hui & Belkin, 2021; Han et al., 2022), and is commonly considered in NC analyses (Han et al., 2022; Tirer & Bruna, 2022).

• **Pre- and Post-activation Kernels:** For any two inputs $\mathbf{x}^{c,i}, \mathbf{x}^{c',j} \in \mathbb{R}^{d_0}$, we denote their corresponding pre- and post-activation features as $\mathbf{z}^{c,i}, \mathbf{z}^{c',j} \in \mathbb{R}^{d_1}$ and $\phi(\mathbf{z}^{c,i}), \phi(\mathbf{z}^{c',j}) \in \mathbb{R}^{d_1}$, respectively. The pre and post-activation kernels corresponding to layer $l = 1$ are given by:

$$\begin{aligned} K^{(1)}(\mathbf{x}^{c,i}, \mathbf{x}^{c',j}) &= \mathbf{z}^{c,i\top}\mathbf{z}^{c',j} = \left( \mathbf{b}^{(1)} + \mathbf{W}^{(1)}\mathbf{x}^{c,i} \right)^\top \left( \mathbf{b}^{(1)} + \mathbf{W}^{(1)}\mathbf{x}^{c',j} \right), \\ Q^{(1)}(\mathbf{x}^{c,i}, \mathbf{x}^{c',j}) &= \phi(\mathbf{z}^{c,i})^\top \phi(\mathbf{z}^{c',j}). \end{aligned} \tag{4}$$

## 4 Within-Class Variability Metric (NC1) for Kernels

In this section, we derive an NC1 metric that depends on the features only through the kernel function. Let $\mathbf{H} \in \mathbb{R}^{N \times d}$ be a matrix encapsulating arbitrary feature vectors associated with samples of the $C$ classes. We define the within-class covariance $\mathbf{\Sigma}_W(\mathbf{H})$ and between-class covariance $\mathbf{\Sigma}_B(\mathbf{H})$ matrices of the features as:

$$\mathbf{\Sigma}_W(\mathbf{H}) = \frac{1}{N} \sum_{c=1}^{C} \sum_{i=1}^{n_c} \left( \mathbf{h}^{c,i} - \overline{\mathbf{h}}^c \right) \left( \mathbf{h}^{c,i} - \overline{\mathbf{h}}^c \right)^\top ; \mathbf{\Sigma}_B(\mathbf{H}) = \frac{1}{C} \sum_{c=1}^{C} \left( \overline{\mathbf{h}}^c - \overline{\mathbf{h}}^G \right) \left( \overline{\mathbf{h}}^c - \overline{\mathbf{h}}^G \right)^\top , \tag{5}$$

where $\overline{\mathbf{h}}^c = \frac{1}{n_c} \sum_{i=1}^{n_c} \mathbf{h}^{c,i}, \forall c \in [C]$ and $\overline{\mathbf{h}}^G = \frac{1}{N} \sum_{c=1}^{C} \sum_{i=1}^{n_c} \mathbf{h}^{c,i}$ represent the class mean vectors and the global mean vector, respectively. Additionally, we consider the total covariance $\widetilde{\mathbf{\Sigma}}_T(\mathbf{H})$ and non-centered between-class covariance $\widetilde{\mathbf{\Sigma}}_B(\mathbf{H})$ matrices as follows:

$$\widetilde{\mathbf{\Sigma}}_T(\mathbf{H}) = \frac{1}{N} \sum_{c=1}^{C} \sum_{i=1}^{n_c} \mathbf{h}^{c,i} \mathbf{h}^{c,i\top}, \qquad \widetilde{\mathbf{\Sigma}}_B(\mathbf{H}) = \frac{1}{C} \sum_{c=1}^{C} \overline{\mathbf{h}}^c \overline{\mathbf{h}}^{c\top}. \tag{6}$$

Based on these formulations, we define the variability metric $\mathcal{NC}_1(\mathbf{H})$, introduced in (Tirer et al., 2023) and used also in (Kothapalli et al., 2023; Wang et al., 2023; Yaras et al., 2023), as:

$$\mathcal{NC}_1(\mathbf{H}) := \frac{\text{tr}(\mathbf{\Sigma}_W(\mathbf{H}))}{\text{tr}(\mathbf{\Sigma}_B(\mathbf{H}))}. \tag{7}$$

In the following theorem, we formulate the traces $\text{tr}(\mathbf{\Sigma}_W(\mathbf{H}))$ and $\text{tr}(\mathbf{\Sigma}_B(\mathbf{H}))$ using an arbitrary kernel function $Q : \mathbb{R}^{d_0} \times \mathbb{R}^{d_0} \to \mathbb{R}$ that expresses inner product of data samples in feature space.

**Theorem 4.1.** *For any two data points $\mathbf{x}^{c,i}, \mathbf{x}^{c',j}$, let the inner-product of their associated features $\mathbf{h}^{c,i}, \mathbf{h}^{c',j}$ be given by a kernel $Q : \mathbb{R}^{d_0} \times \mathbb{R}^{d_0} \to \mathbb{R}$ as $Q(\mathbf{x}^{c,i}, \mathbf{x}^{c',j}) = \mathbf{h}^{c,i\top}\mathbf{h}^{c',j}$. The traces of covariance matrices $\text{tr}(\mathbf{\Sigma}_W(\mathbf{H}))$ and $\text{tr}(\mathbf{\Sigma}_B(\mathbf{H}))$ can now be formulated as:*

$$\text{tr}(\mathbf{\Sigma}_W(\mathbf{H})) = \frac{1}{N} \sum_{c=1}^{C} \sum_{i=1}^{n_c} Q(\mathbf{x}^{c,i}, \mathbf{x}^{c,i}) - \frac{1}{C} \sum_{c=1}^{C} \frac{1}{n_c^2} \sum_{i=1}^{n_c} \sum_{j=1}^{n_c} Q(\mathbf{x}^{c,i}, \mathbf{x}^{c,j}), \tag{8}$$

$$\text{tr}(\mathbf{\Sigma}_B(\mathbf{H})) = \frac{1}{C} \sum_{c=1}^{C} \frac{1}{n_c^2} \sum_{i=1}^{n_c} \sum_{j=1}^{n_c} Q(\mathbf{x}^{c,i}, \mathbf{x}^{c,j}) - \frac{1}{N^2} \sum_{c=1}^{C} \sum_{c'=1}^{C} \sum_{i=1}^{n_c} \sum_{j=1}^{n_{c'}} Q(\mathbf{x}^{c,i}, \mathbf{x}^{c',j}). \tag{9}$$

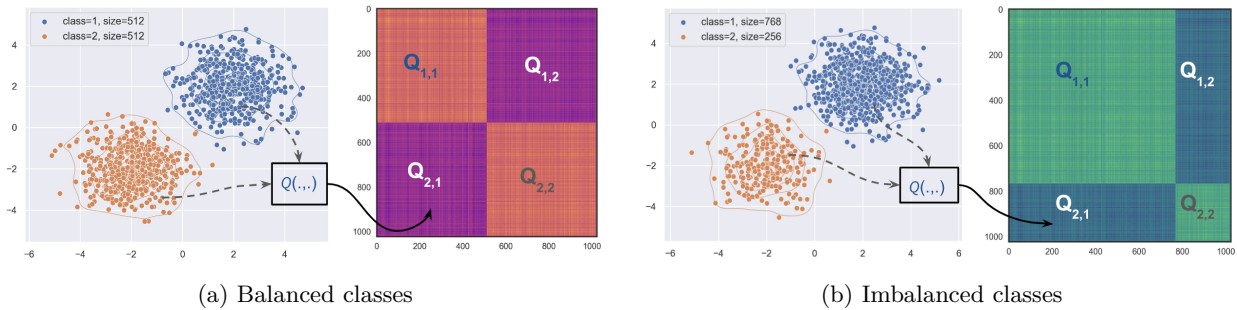

(a) Balanced classes                                        (b) Imbalanced classes

Figure 1: Visualizing the kernel matrix $\mathbf{Q}$ for the limiting NNGP post-activation kernel function $Q_{GP-Erf}$ : $\mathbb{R}^2 \times \mathbb{R}^2 \to \mathbb{R}$. The data is sampled from two Gaussian distributions in (a) balanced and (b) imbalanced fashion to illustrate the structure of the sub-matrices $\mathbf{Q}_{c,c'}, c, c' \in \{1, 2\}$.

The proof (in Appendix A) leverages matrix trace properties and direct expansions of the covariance matrices $\widetilde{\mathbf{\Sigma}}_T(\mathbf{H}), \widetilde{\mathbf{\Sigma}}_B(\mathbf{H}), \mathbf{\Sigma}_W(\mathbf{H})$ in terms of vector outer-products to arrive at the results.

Observe that Theorem 4.1 allows us to replace $Q(\cdot, \cdot)$ with any suitable kernel formulation corresponding to the features. In the following sections, we leverage this flexibility to analyze and compare NC1 for kernels that model the behavior of neural networks.

## 5 Activation Variability of NNGP and NTK

In the UFM-based analysis of NC, the assumption is that the deepest features $\mathbf{H}$ associated with the training samples $\mathbf{X}$ are free optimization variables, thus losing any ability to analyze the effect of the training data, apart from the balancedness of its labels $\mathbf{Y}$. In this section, we address these shortcomings by analyzing the role of data distributions on $\mathcal{NC}_1(\mathbf{H})$ in the infinite width regime, where the NN behavior can be well modeled by NNGP and NTK. In particular, we focus on the case where data is sampled from a mixture of Gaussians and understand the limits of $\mathcal{NC}_1(\mathbf{H})$ reduction.

### 5.1 Limiting NNGP Kernel

Under the NN model and initialization stated in Section 3, as the hidden layer width $d_1 \to \infty$, we can characterize the pre-activation kernel $K^{(1)}(\mathbf{x}^{c,i}, \mathbf{x}^{c',j})$ in terms of the GP limit (Neal, 1995) (commonly referred to as the NNGP limit (Lee et al., 2018)) as follows:

$$K_{GP}^{(1)}(\mathbf{x}^{c,i}, \mathbf{x}^{c',j}) = \sigma_b^2 + \frac{\sigma_w^2}{d_0} \mathbf{x}^{c,i\top} \mathbf{x}^{c',j}. \tag{10}$$

In this limit, the post-activation kernel $Q_{GP}^{(1)}(\cdot, \cdot)$ can have a closed form representation depending on the activation function $\phi(\cdot)$ (Lee et al., 2018; 2019). The expression for Erf activation (Williams, 1996) is:

$$Q_{GP-Erf}^{(1)}(\mathbf{x}^{c,i}, \mathbf{x}^{c',j}) = \frac{2}{\pi} \arcsin\left(\frac{2K_{GP}^{(1)}(\mathbf{x}^{c,i}, \mathbf{x}^{c',j})}{\sqrt{1 + 2K_{GP}^{(1)}(\mathbf{x}^{c,i}, \mathbf{x}^{c,i})}\sqrt{1 + 2K_{GP}^{(1)}(\mathbf{x}^{c',j}, \mathbf{x}^{c',j})}}\right). \tag{11}$$

The formulation for ReLU-based kernel $Q_{GP-ReLU}^{(1)}(\cdot, \cdot)$ (Cho & Saul, 2009) is presented in the Appendix B. Given a kernel function $Q(\cdot, \cdot)$ and samples $\mathbf{X}$, we can formulate the kernel Gram matrix $\mathbf{Q} \in \mathbb{R}^{N \times N}$ as:

$$\mathbf{Q} = \begin{bmatrix} \mathbf{Q}_{1,1} & \cdots & \mathbf{Q}_{1,C} \\ \vdots & \ddots & \vdots \\ \mathbf{Q}_{C,1} & \cdots & \mathbf{Q}_{C,C} \end{bmatrix}_{N \times N}, \mathbf{Q}_{c,c'} = \begin{bmatrix} Q(\mathbf{x}^{c,1}, \mathbf{x}^{c',1}) & \cdots & Q(\mathbf{x}^{c,1}, \mathbf{x}^{c',n_{c'}}) \\ \vdots & \ddots & \vdots \\ Q(\mathbf{x}^{c,n_c}, \mathbf{x}^{c',1}) & \cdots & Q(\mathbf{x}^{c,n_c}, \mathbf{x}^{c',n_{c'}}) \end{bmatrix}_{n_c \times n_{c'}}. \tag{12}$$

Considering the NNGP kernel in (11), we illustrate an example $\mathbf{Q}$ matrix in Figure 1 to visualize the sub-matrices based on the imbalance/balance of class sizes.

## 5.2 Limiting NTK

In the infinite width limit, we also analyze the NTK (Jacot et al., 2018) to understand the effect of optimization on the NN's features in the "lazy regime". Specifically, a well-known result is that in the infinite width limits (with initialization as per Section 3), the deepest feature mapping of the NN is fixed during gradient descent optimization with small enough learning rate, and is characterized by the NTK. Formally, the recursive relationship between the NTK and NNGP (Jacot et al., 2018; Lee et al., 2019) can be given as follows:

$$\Theta_{NTK}^{(2)}(\mathbf{x}^{c,i}, \mathbf{x}^{c',j}) = K_{GP}^{(2)}(\mathbf{x}^{c,i}, \mathbf{x}^{c',j}) + K_{GP}^{(1)}(\mathbf{x}^{c,i}, \mathbf{x}^{c',j})\dot{Q}_{GP}^{(1)}(\mathbf{x}^{c,i}, \mathbf{x}^{c',j}). \tag{13}$$

Here, $K_{GP}^{(2)}(\mathbf{x}^{c,i}, \mathbf{x}^{c',j})$ can be defined using the recursive formulation (Lee et al., 2018; Jacot et al., 2018):

$$K_{GP}^{(2)}(\mathbf{x}^{c,i}, \mathbf{x}^{c',j}) = \sigma_b^2 + \sigma_w^2 Q_{GP}^{(1)}(\mathbf{x}^{c,i}, \mathbf{x}^{c',j}). \tag{14}$$

Similar to the activation function specific formulations of $Q_{GP}^{(1)}(\cdot, \cdot)$ in (11), we define the `Erf` based derivative kernel $\dot{Q}_{GP-Erf}^{(1)}(\cdot, \cdot)$ as follows:

$$\dot{Q}_{GP-Erf}^{(1)}(\mathbf{x}^{c,i}, \mathbf{x}^{c',j}) = \frac{4}{\pi} \det\left(\begin{bmatrix} 1 + 2K_{GP}^{(1)}(\mathbf{x}^{c,i}, \mathbf{x}^{c,i}) & 2K_{GP}^{(1)}(\mathbf{x}^{c,i}, \mathbf{x}^{c',j}) \\ 2K_{GP}^{(1)}(\mathbf{x}^{c',j}, \mathbf{x}^{c,i}) & 1 + 2K_{GP}^{(1)}(\mathbf{x}^{c',j}, \mathbf{x}^{c',j}) \end{bmatrix}\right)^{-1/2}. \tag{15}$$

The formulation for `ReLU`-based kernel $\dot{Q}_{GP-ReLU}^{(1)}(\mathbf{x}^{c,i}, \mathbf{x}^{c',j})(\cdot, \cdot)$ is presented in the Appendix B.

**Remark on the limiting NTK.** Denote by $w, \phi$ the parameters and the activation function of an $L$-layer NN $\psi(\cdot)$. The limiting NNGP is defined directly on the product of neurons, $\mathbb{E}_w\langle\phi(z_i(\mathbf{x})), \phi(z_i(\tilde{\mathbf{x}}))\rangle$, and provides kernel expressions for the inner product of features at different layers of the network (via recursion similar to (11)). On the other hand, the limiting NTK is defined as the inner product of the output's gradients $\mathbb{E}_w\langle\nabla_w\psi(\mathbf{x}), \nabla_w\psi(\tilde{\mathbf{x}})\rangle$. The NTK theory shows that this can model the inner product only of the deepest features (i.e., the output of the penultimate layer)[1]. Yet, NC particularly considers the deepest features.

## 5.3 On the Limitations of NTK to Exhibit 'More' Collapsed Features than NNGP

Notice that even for shallow NNs, it is challenging to theoretically analyze NNGP and NTK for general data. Therefore, we consider a simplified setting to analyze the NC1 properties of these kernels. Formally, consider a 1-dimensional Gaussian dataset (i.e., $d_0 = 1$) with $C = 2$ classes. The data points $\{x^{1,i}\}, \forall i \in [n_1]$ belonging to class $c = 1$ are independently sampled from $\mathcal{N}(\mu_1, \sigma_1^2)$ and have the labels $y^{1,i} = -1, \forall i \in [n_1]$. Similarly, the data points $\{x^{2,j}\}, \forall j \in [n_2]$ belonging to class $c = 2$ are independently sampled from $\mathcal{N}(\mu_2, \sigma_2^2)$ and have the labels $y^{2,j} = 1, \forall j \in [n_2]$.

• **Assumption 1:** For $\mu_1 < 0, \mu_2 > 0$, let $\sigma_1, \sigma_2 > 0$ be small enough such that $|\mu_1| \gg \sigma_1$, $|\mu_2| \gg \sigma_2$ and $\forall i \in [n_1], j \in [n_2], x^{1,i}x^{2,j} < 0$ almost surely.

• **Assumption 2:** The dataset $\mathbf{X} \in \mathbb{R}^{N \times 1}$ consists of large enough samples $n_1, n_2 \gg 1$.

• **Assumption 3:** The 2L-FCN $\psi(\cdot)$ has output layer dimension $d_2 = 1$ and $\sigma_b \to 0$.

These assumptions present a scenario where the samples of the two classes are sufficiently far from the origin in opposite directions. Thus, the simplest prediction rule pertains to the sign of a sample.

**Theorem 5.1** (`ReLU` Activation)**.** *Under Assumptions 1-3, let $\phi(\cdot)$ be the ReLU activation. Denote by $\mathbf{H}_{GP}, \mathbf{H}_{NTK}$ the features associated with NNGP $Q_{GP-ReLU}^{(1)}$ and NTK $\Theta_{NTK-ReLU}^{(2)}$, respectively. Then:*

$$\mathbb{E}\left[\mathcal{NC}_1(\mathbf{H}_{GP})\right] = \mathbb{E}\left[\mathcal{NC}_1(\mathbf{H}_{NTK})\right] = \frac{\sum_{c=1}^2 \frac{n_c\mu_c^2 + n_c\sigma_c^2}{N} - \frac{\mu_c^2}{2}}{\left(\sum_{c=1}^2 \frac{\mu_c^2}{2} - \frac{n_c^2\mu_c^2}{N^2}\right) - \frac{2}{N^2}\prod_{c=1}^2 n_c\mu_c} + \Delta_{h.o.t} \tag{16}$$

*where $\Delta_{h.o.t}$ is a term that vanishes as $\{n_c\}$ increase.*

---

[1]Note that we denote NNGP ($Q_{GP}^{(1)}(\mathbf{x}^{c,i}, \mathbf{x}^{c',j})$) and NTK ($\Theta_{NTK}^{(2)}(\mathbf{x}^{c,i}, \mathbf{x}^{c',j})$) with different superscripts to be consistent with the literature. Yet both are associated with the output of the single hidden layer.

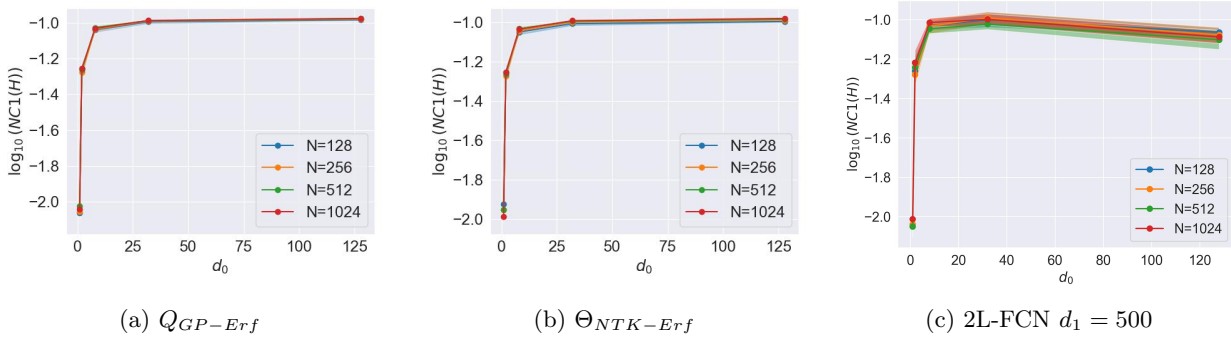

(a) $Q_{GP-Erf}$        (b) $\Theta_{NTK-Erf}$        (c) 2L-FCN $d_1 = 500$

Figure 2: $\mathcal{NC}_1(\mathbf{H})$ of (a) the post-activation NNGP kernel ($Q^{(1)}_{GP-Erf}$), (b) NTK ($\Theta^{(2)}_{NTK-Erf}$), (c) 2L-FCN with $d_1 = 500$ for `Erf` activation on dataset $\mathcal{D}_1(N, d_0)$ (as per (17)).

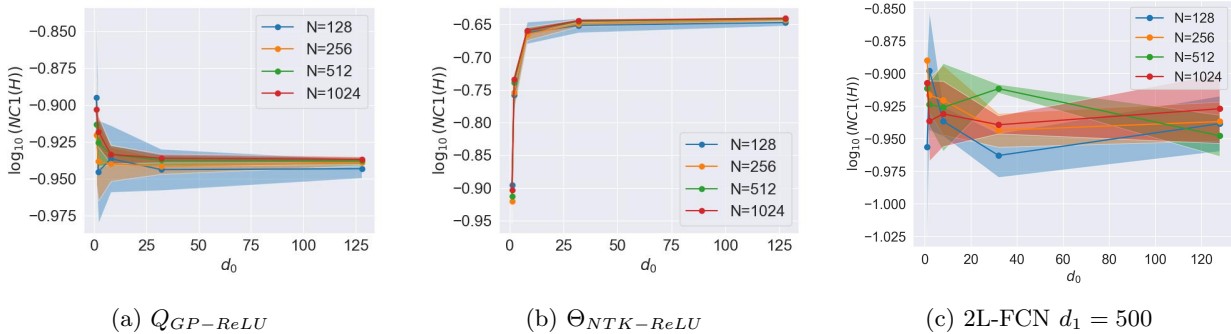

(a) $Q_{GP-ReLU}$        (b) $\Theta_{NTK-ReLU}$        (c) 2L-FCN $d_1 = 500$

Figure 3: $\mathcal{NC}_1(\mathbf{H})$ of (a) the post-activation NNGP kernel ($Q^{(1)}_{GP-ReLU}$), (b) NTK ($\Theta^{(2)}_{NTK-ReLU}$), (c) 2L-FCN with $d_1 = 500$ for `ReLU` activation on dataset $\mathcal{D}_1(N, d_0)$ (as per (17)).

Appendix D presents the proof by calculating the expected values of $Q^{(1)}_{GP-ReLU}(\mathbf{x}^{c,i}, \mathbf{x}^{c',j})$ and employing Theorem 4.1. For a better understanding of the result, consider the balanced class scenario with $n_1 = n_2 = N/2$. This gives us: $\mathbb{E}\left[\mathcal{NC}_1(\mathbf{H}_{GP/NTK})\right] = 2(\sigma_1^2 + \sigma_2^2)/(\mu_1 - \mu_2)^2 + \Delta_{h.o.t}$, which captures the sum of the within-class variance of $\mathbf{X}$ in the numerator and the between-class variance in the denominator of the first term.

**The `Erf` activation case.** We present a similar analysis with the NNGP and NTK with `Erf` activation in Appendix E (as the terms involved in the formulation are relatively complex than the `ReLU` case). In the case of `ReLU` with balanced data, observe that the numerator (corresponding to $\Sigma_W(\mathbf{H}_{GP/NTK})$) solely depends on $\sigma_c^2$, however, for `Erf` activation, our analysis shows a dependence on terms $\propto \sigma_c^2 \mu_c^{-6}$ i.e, it depends on inverse of higher powers of class means $\mu_c$ as well (see (92) in Appendix E). Similar analysis for $\Sigma_B(\mathbf{H}_{GP/NTK})$ with `Erf` in (100) shows a dependence on terms $\propto \sigma_c^2 \mu_c^{-4}$. Importantly, under Assumptions 1-3, we show similar values of the expected NC1 metric for NNGP and NTK even for the `Erf` activation.

**Key Takeaway.** Our results show that even with 1-D Gaussian data, the expected $\mathcal{NC}_1(\mathbf{H})$ of the NTK closely approximates the NNGP counterparts. Perhaps surprisingly, this shows that NTK does not represent more collapsed features than NNGP, despite being associated with NN gradient-based optimization. Namely, we have established another result that shows that training in the lazy regime provably deviates from the practical feature learning of NNs (Woodworth et al., 2020; Ghorbani et al., 2020; Wei et al., 2019; Yehudai & Shamir, 2019; Li et al., 2020).

### 5.4 Experiments with High Dimensional Data

**Setup.** We conduct experiments on datasets with varying sample sizes and input dimensions to verify our theoretical results and show that insights generalize (e.g., beyond $d_0 = 1$). For $C = 2$, a dataset size $N$ chosen from $\{128, 256, 512, 1024\}$, and input dimension $d_0$ chosen from $\{1, 2, 8, 32, 128\}$, we create the data vector

and label pairs as follows:

$$\begin{aligned}
\mathcal{D}_1(N, d_0) = &\left\{ (\mathbf{x}^{1,i} \sim \mathcal{N}(-2 * \mathbf{1}_{d_0}, 0.25 * \mathbf{I}_{d_0}), y^{1,i} = -1), \forall i \in [N/2]) \right\} \\
& \cup \left\{ (\mathbf{x}^{2,j} \sim \mathcal{N}(2 * \mathbf{1}_{d_0}, 0.25 * \mathbf{I}_{d_0}), y^{1,i} = 1), \forall j \in [N/2]) \right\}.
\end{aligned} \tag{17}$$

The vectors and labels from the dataset can then be arranged into the matrix form (as described in the setup) for analysis. The sampling procedure is repeated 10 times for each $(N, d_0)$.

**Erf activation leads to more collapsed features than `ReLU`.** For the low-dimensional case of $d_0 = 1$ and `Erf` activation, observe from Figure 2a that $\mathcal{NC}_1(\mathbf{H})$ has a small value of $\approx 10^{-2.1}$. On the contrary, Figure 3a illustrates that for `ReLU`, $\mathcal{NC}_1(\mathbf{H})$ is more than an order of magnitude larger ($\approx 10^{-0.95}$) than the former. Furthermore, Figures 2b and 3b corresponding to NTK (with `Erf` and `ReLU` respectively) do not exhibit significantly different values from the NNGP counterparts. These observations empirically verify our theoretical results above for $d = 1$. Furthermore, notice that the $\mathcal{NC}_1(\mathbf{H})$ values are higher in the case of `ReLU` than `Erf` across all the dimensions. We also observe such behavior when training the 2L-FCN networks with hidden layer width $d_1 = 500$ (see Appendix G for training details). When $d_1 = 500$, the finite width effects of training come into play (see following sections) and we observe a shift in the trend of $\mathcal{NC}_1(\mathbf{H})$ for $d_0 > 32$. Nonetheless, the `Erf` case in Figure 2c still exhibits more collapse than `ReLU` case in Figure 3c.

**NTK does not result in more collapsed features than NNGP in higher dimensions.** As the data dimension $d_0$ increases, $\mathcal{NC}_1(\mathbf{H})$ increases at similar rates for $Q_{GP-Erf}$ and $\Theta_{NTK-Erf}$. With `ReLU` activation, $\mathcal{NC}_1(\mathbf{H})$ remains almost constant for $Q_{GP-ReLU}$ and exhibits an increasing trend for $\Theta_{NTK-ReLU}$ — *implying less collapse for NTK*. All these observations corroborate our theory on the limitation of analyzing NC with NTK. We present additional experimental results for imbalanced datasets in Appendix G and show that for a given $N$, the trends of $\mathcal{NC}_1(\mathbf{H})$ for increasing $d_0$ can vary based on the imbalance ratio of classes (i.e., $n_1/n_2$). These observations provide zero-order reasoning for NN behavior where the feature mapping is learned based on data properties such as dimension.

## 6 Activation Variability of *data-aware* GP Kernels

The explicit kernel formulations in the infinite width limit ($d_1 \to \infty$) have allowed us to go beyond the unconstrained features assumption and preserve a link between the features and the data. Yet, the NC phenomenon relates to NN training, while NNGP relates to NN at initialization and NTK has been found unsuitable for NC analysis. Thus, we wish to contrast NNGP with a kernel that is an alternative to NTK and takes into account both optimization and data. To this end, we transition to a large but finite width ($d_1 \gg 1$) and large sample ($N \gg 1$) setting and analyze the *data-aware* GP kernels by (Seroussi et al., 2023) for FCNs.

### 6.1 Equations of State (EoS) for the *data-aware* GP Kernel

A transition from the infinite to finite width regime can introduce various "corrections" to the pre-and post-activations of a $L$-layer FCN, which account for the different behavior. In this context, (Seroussi et al., 2023) have observed the following dominant corrections: (1) The mean and covariance of the pre-activations deviate from that of a random FCN in a nearly Gaussian manner and, (2) the collective effect of activations from the $(l+1)^{th}$ and $(l-1)^{th}$ layers determine the covariance of activations in the $l^{th}$ layer. Therefore, while for infinite-width DNNs, the kernels representations are inert, for finite DNNs they adapt to the data and yield a tractable data-aware Gaussian Process. Specifically, based on the aforementioned observations, (Seroussi et al., 2023) employ a Variational Gaussian Approximation (VGA) approach and derived the following system of equations for the pre and post-activation kernels $K^{(l)}(\cdot, \cdot), Q^{(l)}(\cdot, \cdot), l \in [L]$ respectively. The solution of this system models the state of the DNN at convergence (after the training phase), and has been empirically shown to yield accurate predictions in various settings. We formally define the EoS for a 2-layer FCN as follows (based on specializing the generic $L$-layer formulation in equation 5 in (Seroussi et al., 2023) to $L = 2$, as done in equation 95 in their arxiv extended version):

**Definition 6.1.** *The "Equations of State" (EoS) for pre and post-activation kernels of a 2-layer FCN with* `Erf` *activation, no bias, and $d_2 = 1$ are given by:*

$$\bar{\mathbf{f}} = \mathbf{Q}^{(1)}[\sigma^2 \mathbf{I} + \mathbf{Q}^{(1)}]^{-1}\mathbf{y}$$

$$[\mathbf{Q}^{(1)}]_{ij} = \sigma_a^2 \frac{2}{\pi} \arcsin\left(2K_{ij}^{(1)} \cdot \left(\sqrt{1 + 2K_{ii}^{(1)}}\sqrt{1 + 2K_{jj}^{(1)}}\right)^{-1}\right)$$

$$[\mathbf{C}^{-1}]_{ij} = \frac{d_0}{\sigma_w^2}\delta_{ij} + \frac{1}{d_1}\text{tr}\left\{\mathbf{A}^{(1)}\partial_{C_{ij}}\mathbf{Q}^{(1)}\right\} \tag{18}$$

$$\mathbf{A}^{(1)} = -(\mathbf{y} - \bar{\mathbf{f}})(\mathbf{y} - \bar{\mathbf{f}})^\top \sigma^{-4} + [\mathbf{Q}^{(1)} + \sigma^2 \mathbf{I}]^{-1}$$

*Here, $\mathbf{C} \in \mathbb{R}^{d_0 \times d_0}$ models the statistical covariance of a row of $\mathbf{W}^{(1)}$, initialized with $(\sigma_w^2/d_0)\mathbf{I}$, $\mathbf{K}^{(1)} = \mathbf{X}^\top \mathbf{C} \mathbf{X} \in \mathbb{R}^{N \times N}$, $\sigma > 0$ is the regularization parameter, and $\bar{\mathbf{f}} \in \mathbb{R}^N$ corresponds to the prediction of the 2-layer FCN (governed by the EoS). Additionally, $\mathbf{K}^{(1)}, \mathbf{Q}^{(1)} \in \mathbb{R}^{N \times N}$ are the kernel matrices associated with kernel functions $K^{(1)}(\cdot, \cdot), Q^{(1)}(\cdot, \cdot)$.*

• **Relationship with NNGP.** At initialization, we set $\mathbf{C} = (\sigma_w^2/d_0)\mathbf{I}$, which implies $\mathbf{K}^{(1)} = (\sigma_w^2/d_0)\mathbf{X}^\top \mathbf{X}$. This resulting $\mathbf{K}^{(1)}$ exactly matches the kernel matrix for the pre-activation GP kernel (10) $K_{GP}^{(1)}(\mathbf{x}^{c,i}, \mathbf{x}^{c',j}) = (\sigma_w^2/d_0)\mathbf{x}^{c,i\top}\mathbf{x}^{c',j}$ as $\sigma_b \to 0$. Similarly, the matrix $\mathbf{Q}^{(1)}$ corresponds to the $Q_{GP-Erf}(\cdot, \cdot)$ kernel function defined in (11). The EoS provides a mechanism for transitioning from NNGP kernels to finite-width-based kernels that adapt to the data. Intuitively, observe that the predictions $\bar{\mathbf{f}}$ are formulated based on kernel ridge regression with $\mathbf{Q}^{(1)}$ and $\mathbf{y}$. The $\mathbf{Q}^{(1)}$ matrix along with $\mathbf{y}$ and the initial predictions $\bar{\mathbf{f}}$ are then used to update the weight covariance matrix $\mathbf{C}$. Here every entry $[\mathbf{C}^{-1}]_{ij}$ involves a trace operation on a matrix product, resulting in a weighted sum across entries of $\partial_{C_{ij}}\mathbf{Q}^{(1)}$ (i.e the $N^2$ pairs of data samples).

## 6.2 Numerical Solutions of EoS

We solve the EoS using the Newton-Krylov method with an annealing schedule (as originally proposed by (Seroussi et al., 2023)) using the `scipy.optimize.newton_krylov` python API. We initialize $\mathbf{C}$ with the GP limit value of $(\sigma_w^2/d_0)\mathbf{I}_{d_0}$ and choose a large annealing factor (ex: $10^5$) as the value for $d_1$. The result of optimizing with `newton_krylov` is a new $\mathbf{C}$, which in addition to a lower annealing factor is used as an input for the next `newton_krylov` function call. This loop is repeated until the end of an annealing schedule. For instance, to analyze the EoS corresponding to $d_1 = 500$, we choose the following step-wise annealing factors:

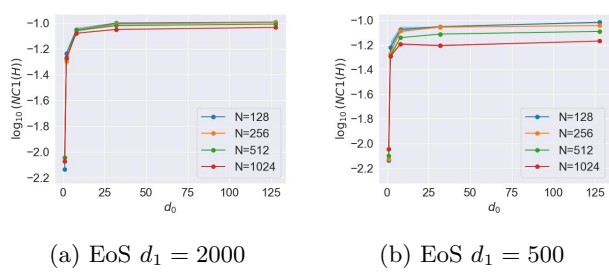

(a) EoS $d_1 = 2000$       (b) EoS $d_1 = 500$

Figure 4: $\mathcal{NC}_1(\mathbf{H})$ of the $\mathbf{Q}^{(1)}$ kernel obtained by solving the EoS (a) $d_1 = 2000$ (b) $d_1 = 500$ on $\mathcal{D}_1(N, d_0)$.

$$\texttt{factors} = [\underbrace{10^5, 9*10^4, \cdots, 2*10^4}_{\texttt{step}=-10^4}, \underbrace{10^4, 9*10^3, \cdots, 2*10^3}_{\texttt{step}=-10^3}, \underbrace{10^3, \cdots, 500}_{\texttt{step}=-10^2}]. \tag{19}$$

Similarly, for a choice of $d_1 = 2000$, we select the slice of the above list up to 2000. Selecting the schedule is a manual operation and can be treated as a hyper-parameter. In our experiments, we observed that this schedule is sufficient to obtain insights on the NC1 metrics of $\mathbf{Q}^{(1)}$. Thus, we leave the exploration of various annealing strategies as future work.

## 6.3 Experiments

**Setup:** We train a 2L-FCN with $d_1 = 500, \sigma_w = 1, \sigma_b = 0$ and `Erf` activation using (vanilla) Gradient Descent with a learning rate of $10^{-3}$ and weight-decay $10^{-6}$ for 1000 steps. To numerically solve the EoS, we employ the approach described in Section 6.2, with the final annealing factor of $d_1 = 500$ and $\sigma_a^2 = 1/128$ (as per the critical scaling value suggested in (Seroussi et al., 2023)).

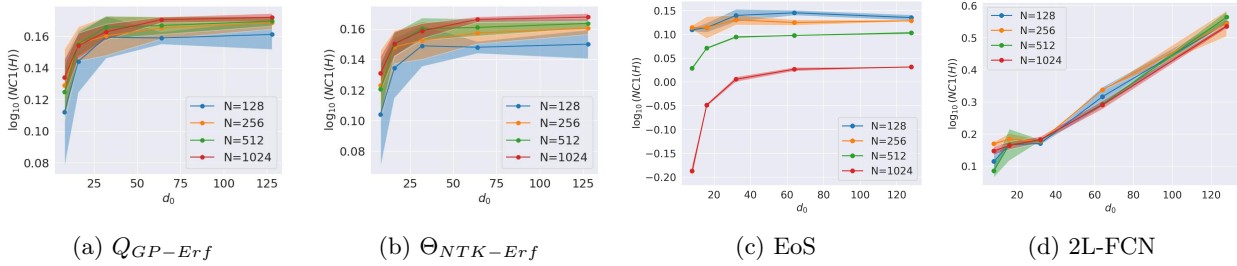

(a) $Q_{GP-Erf}$      (b) $\Theta_{NTK-Erf}$      (c) EoS      (d) 2L-FCN

Figure 5: $\mathcal{NC}_1(\mathbf{H})$ of the limiting kernels, adaptive kernel (EoS) with final annealing factor $d_1 = 500$ and 2L-FCN with $d_1 = 500$ and `Erf` activation. The dimension $d_0$ on the x-axis is chosen from $\{8, 16, 32, 64, 128\}$. For a particular $N$, we sample the vectors $\mathbf{x}^{1,i} \sim \mathcal{N}(-2 * \mathbf{1}_{d_0}, 4 * \mathbf{I}_{d_0}), y^{1,i} = -1, i \in [N/2]$ for class 1 and $\mathbf{x}^{2,j} \sim \mathcal{N}(2 * \mathbf{1}_{d_0}, 4 * \mathbf{I}_{d_0}), y^{2,j} = 1, j \in [N/2]$ for class 2.

**NC1 of EoS on linearly separable datasets.** We solve the EoS (initialized with $\mathbf{C} = (\sigma_w^2/d_0)\mathbf{I}_{d_0}$) and obtain the stable state using the Newton-Krylov method with an annealing schedule, as described above. At initialization, $\mathbf{Q}^{(1)}$ is exactly described by the limiting NNGP kernel matrix, which has been analyzed in the previous section. Now, by solving the EoS with the final annealing factors as 2000 and 500 (which correspond to a 2L-FCN with hidden layer widths $d_1 = 2000, 500$ respectively), we illustrate the NC1 metrics of $\mathbf{Q}^{(1)}$ in Figure 4 for the running example of parameterized balanced 2 class datasets $\mathcal{D}_1(N, d_0)$ (17). Notice that for $d_1 = 2000$, the $\mathcal{NC}_1(\mathbf{H})$ values for $\mathbf{Q}^{(1)}$ in Figure 4a closely resemble the plots for the limiting NNGP kernel $Q_{GP-Erf}$ in Figure 2a and the NTK in Figure 2b. However, we observe noticeable changes in the metrics when $d_1 = 500$. Especially, for $d_0 \geq 8$ and $N \geq 512$, the $\mathcal{NC}_1(\mathbf{H})$ values for the EoS in Figure 4b exhibit a noticeable reduction compared to $Q_{GP-Erf}$ and $\Theta_{NTK-Erf}$. Furthermore, notice that such a reduction aligns with the behavior of 2L-FCN in Figure 2c and reflects the departure of $\mathbf{Q}^{(1)}$ in EoS from the initial NNGP state to a feature learning state.

**NC1 of EoS on non-linearly separable datasets.** Unlike the separable Gaussian datasets in the above sections, the real-world datasets are relatively more complex. To simulate such a scenario, we sample $\mathbf{x}^{1,i} \sim \mathcal{N}(-2 * \mathbf{1}_{d_0}, 4 * \mathbf{I}_{d_0}), y^{1,i} = -1, i \in [N/2]$ for class 1 and $\mathbf{x}^{2,j} \sim \mathcal{N}(2 * \mathbf{1}_{d_0}, 4 * \mathbf{I}_{d_0}), y^{2,j} = 1, j \in [N/2]$ for class 2 of our dataset. Essentially, these are scenarios where there is a significant overlap between samples of the two classes. First, we note that we had to increase the learning rate of our 2L-FCN from $10^{-3}$ to $5 \cdot 10^{-3}$ and run GD for 2000 epochs for convergence. For dimensions $d_0 = \{8, 16, 32\}$, the EoS reasonably approximates the 2L-FCN but for $d_0 = \{64, 128\}$, the $\mathcal{NC}_1(\mathbf{H})$ values for 2L-FCN turned out to be almost twice as large as the EoS (see Figure 5). We also present additional experiments with more number of classes, imbalanced classes and deeper FCN networks in Appendix G.

## 7 Limitations

We highlight the difficulties in the theoretical/empirical analysis of NC1 with EoS. The primary bottleneck is a lack of rigorous study on the existence and uniqueness of solutions (As also highlighted by (Seroussi et al., 2023)). Since we deviate from the lazy regime and deal with kernels in the feature learning setup, we cannot expect simpler closed-form solutions like the limiting NNGP/NTK for the EoS. However, numerical solutions to the EoS can sometimes be time-consuming and require a manual selection of the annealing schedule. This is a tradeoff that can be improved with future research. Furthermore, the role of scaling $N, d_0, d_1$ on NC1 is yet to be fully understood and we hope that our analysis lays the groundwork for such efforts.

**Remark.** Finally, we point the reader to Appendix F for a discussion on a relative NC1 metric that explicitly incorporates the variability collapse of the data vectors into the NC1 metric. In particular, we aim to differentiate between settings where the neural network learned meaningful features and learned to classify complex datasets or was simply able to leverage the already collapsed data vectors. Our results showcase that in higher dimensions, the data vectors are 'more' collapsed than the activations themselves. Thus showcasing the limitations of the current NC1 metrics and encouraging the reader to explore much richer variants.

# 8 Conclusion

In this paper, we explored whether kernel-based approaches can help understand the role of data in the emergence of the NC phenomenon. By considering a general kernel function, we first formulated the trace expressions for the variability collapse (NC1) of the features of the data samples. By leveraging these results, we provided theoretical and empirical results to showcase that the NTK does not represent more collapsed features than the NNGP for various Gaussian datasets. During this investigation we showed how the choice of the nonlinear activation function affects NC1. Next, to capture the feature-learning aspects of finite-width neural networks, we switched to a *data-aware* GP kernel approach whose state equations (EoS) facilitate the transition of the post-activation kernel beyond the NNGP limit. We empirically showed that it yields lower NC1 than NNGP but may not be aligned with trends of FCNs. The key message of the paper is that advancements in modeling NNs by *adaptive* kernels can benefit NC analysis and lead to a considerable leap compared to current UFM based theory.

# Acknowledgment

TT was supported by ISF grant No. 1940/23 and MOST grant No. 0007091. The authors also thank the anonymous reviewers for their valuable feedback.

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

## A    Proof of Theorem 4.1

To obtain the NC1 formulation corresponding to an arbitrary feature matrix $\mathbf{H}$, we start with a simple relationship between $\widetilde{\boldsymbol{\Sigma}}_T(\mathbf{H}), \widetilde{\boldsymbol{\Sigma}}_B(\mathbf{H}), \boldsymbol{\Sigma}_W(\mathbf{H})$ as follows:

$$
\begin{aligned}
\widetilde{\boldsymbol{\Sigma}}_T(\mathbf{H}) &= \boldsymbol{\Sigma}_W(\mathbf{H}) + \widetilde{\boldsymbol{\Sigma}}_B(\mathbf{H}) \\
\implies \operatorname{tr}\left(\boldsymbol{\Sigma}_W(\mathbf{H})\right) &= \operatorname{tr}\left(\widetilde{\boldsymbol{\Sigma}}_T(\mathbf{H})\right) - \operatorname{tr}\left(\widetilde{\boldsymbol{\Sigma}}_B(\mathbf{H})\right).
\end{aligned}
\tag{20}
$$

Similarly, by considering $\boldsymbol{\Sigma}_G(\mathbf{H}) = \overline{\mathbf{h}}^G \overline{\mathbf{h}}^{G\top}$, we get:

$$
\begin{aligned}
\boldsymbol{\Sigma}_B(\mathbf{H}) &= \widetilde{\boldsymbol{\Sigma}}_B(\mathbf{H}) - \boldsymbol{\Sigma}_G(\mathbf{H}) \\
\implies \operatorname{tr}\left(\boldsymbol{\Sigma}_B(\mathbf{H})\right) &= \operatorname{tr}\left(\widetilde{\boldsymbol{\Sigma}}_B(\mathbf{H})\right) - \operatorname{tr}\left(\boldsymbol{\Sigma}_G(\mathbf{H})\right).
\end{aligned}
\tag{21}
$$

● **Formulating** $\operatorname{tr}\left(\widetilde{\boldsymbol{\Sigma}}_T(\mathbf{H})\right)$**:** Expanding $\widetilde{\boldsymbol{\Sigma}}_T(\mathbf{H})$ into individual outer-products of vectors and leveraging the trace properties leads to the following:

$$
\begin{aligned}
\operatorname{tr}\left(\widetilde{\boldsymbol{\Sigma}}_T(\mathbf{H})\right) &= \operatorname{tr}\left(\frac{1}{N}\sum_{c=1}^{C}\sum_{i=1}^{n_c}\mathbf{h}^{c,i}\mathbf{h}^{c,i\top}\right) = \frac{1}{N}\sum_{c=1}^{C}\sum_{i=1}^{n_c}\operatorname{tr}\left(\mathbf{h}^{c,i}\mathbf{h}^{c,i\top}\right) \\
&= \frac{1}{N}\sum_{c=1}^{C}\sum_{i=1}^{n_c}\operatorname{tr}\left(\mathbf{h}^{c,i\top}\mathbf{h}^{c,i}\right) \\
&= \frac{1}{N}\sum_{c=1}^{C}\sum_{i=1}^{n_c}Q(\mathbf{x}^{c,i}, \mathbf{x}^{c,i})
\end{aligned}
$$

● **Formulating** $\operatorname{tr}\left(\widetilde{\boldsymbol{\Sigma}}_B(\mathbf{H})\right)$**:** Similar to the above analysis, we can reformulate the trace of non-centered between-class covariance matrix $\widetilde{\boldsymbol{\Sigma}}_B(\mathbf{H})$ as:

$$
\begin{aligned}
\operatorname{tr}(\widetilde{\boldsymbol{\Sigma}}_B) &= \operatorname{tr}\left(\frac{1}{C}\sum_{c=1}^{C}\overline{\mathbf{h}}^c\overline{\mathbf{h}}^{c\top}\right) = \frac{1}{C}\sum_{c=1}^{C}\operatorname{tr}\left(\overline{\mathbf{h}}^c\overline{\mathbf{h}}^{c\top}\right) = \frac{1}{C}\sum_{c=1}^{C}\operatorname{tr}\left(\overline{\mathbf{h}}^{c\top}\overline{\mathbf{h}}^c\right) \\
&= \frac{1}{C}\sum_{c=1}^{C}\operatorname{tr}\left(\left[\frac{1}{n_c}\sum_{i=1}^{n_c}\mathbf{h}^{c,i}\right]^{\top}\left[\frac{1}{n_c}\sum_{i=1}^{n_c}\mathbf{h}^{c,i}\right]\right) \\
&= \frac{1}{C}\sum_{c=1}^{C}\frac{1}{n_c^2}\operatorname{tr}\left(\sum_{i=1}^{n_c}\sum_{j=1}^{n_c}\mathbf{h}^{c,i\top}\mathbf{h}^{c,j}\right) = \frac{1}{C}\sum_{c=1}^{C}\frac{1}{n_c^2}\sum_{i=1}^{n_c}\sum_{j=1}^{n_c}\operatorname{tr}\left(\mathbf{h}^{c,i\top}\mathbf{h}^{c,j}\right) \\
&= \frac{1}{C}\sum_{c=1}^{C}\frac{1}{n_c^2}\sum_{i=1}^{n_c}\sum_{j=1}^{n_c}Q(\mathbf{x}^{c,i}, \mathbf{x}^{c,j})
\end{aligned}
$$

- **Formulating** $\text{tr}\left(\boldsymbol{\Sigma}_G(\mathbf{H})\right)$**:** Reformulation of $\text{tr}\left(\boldsymbol{\Sigma}_G(\mathbf{H})\right)$ can be approached along the same lines:

$$
\begin{aligned}
\text{tr}\left(\boldsymbol{\Sigma}_G(\mathbf{H})\right) &= \text{tr}\left(\overline{\mathbf{h}}^G \overline{\mathbf{h}}^{G\top}\right) = \text{tr}\left(\overline{\mathbf{h}}^{G\top} \overline{\mathbf{h}}^G\right) \\
&= \text{tr}\left(\left[\frac{1}{N}\sum_{c=1}^{C}\sum_{i=1}^{n_c}\mathbf{h}^{c,i}\right]^{\top}\left[\frac{1}{N}\sum_{c=1}^{C}\sum_{j=1}^{n_c}\mathbf{h}^{c,j}\right]\right) \\
&= \frac{1}{N^2}\text{tr}\left(\sum_{c=1}^{C}\sum_{i=1}^{n_c}\sum_{c'=1}^{C}\sum_{j=1}^{n_{c'}}\mathbf{h}^{c,i\top}\mathbf{h}^{c',j}\right) = \frac{1}{N^2}\sum_{c=1}^{C}\sum_{i=1}^{n_c}\sum_{c'=1}^{C}\sum_{j=1}^{n_{c'}}\text{tr}\left(\mathbf{h}^{c,i\top}\mathbf{h}^{c',j}\right) \\
&= \frac{1}{N^2}\sum_{c=1}^{C}\sum_{c'=1}^{C}\sum_{i=1}^{n_c}\sum_{j=1}^{n_{c'}}Q(\mathbf{x}^{c,i},\mathbf{x}^{c',j})
\end{aligned}
$$

By using these intermediate results, we can formulate $\text{tr}\left(\boldsymbol{\Sigma}_W(\mathbf{H})\right), \text{tr}\left(\boldsymbol{\Sigma}_B(\mathbf{H})\right)$ as:

$$
\begin{aligned}
\text{tr}(\boldsymbol{\Sigma}_W(\mathbf{H})) &= \text{tr}(\widetilde{\boldsymbol{\Sigma}}_T(\mathbf{H})) - \text{tr}(\widetilde{\boldsymbol{\Sigma}}_B(\mathbf{H})) \\
&= \frac{1}{N}\sum_{c=1}^{C}\sum_{i=1}^{n_c}Q(\mathbf{x}^{c,i},\mathbf{x}^{c,i}) - \frac{1}{C}\sum_{c=1}^{C}\frac{1}{n_c^2}\sum_{i=1}^{n_c}\sum_{j=1}^{n_c}Q(\mathbf{x}^{c,i},\mathbf{x}^{c,j}) \\
\text{tr}(\boldsymbol{\Sigma}_B(\mathbf{H})) &= \text{tr}(\widetilde{\boldsymbol{\Sigma}}_B(\mathbf{H})) - \text{tr}(\boldsymbol{\Sigma}_G(\mathbf{H}))) \\
&= \frac{1}{C}\sum_{c=1}^{C}\frac{1}{n_c^2}\sum_{i=1}^{n_c}\sum_{j=1}^{n_c}Q(\mathbf{x}^{c,i},\mathbf{x}^{c,j}) - \frac{1}{N^2}\sum_{c=1}^{C}\sum_{c'=1}^{C}\sum_{i=1}^{n_c}\sum_{j=1}^{n_{c'}}Q(\mathbf{x}^{c,i},\mathbf{x}^{c',j}).
\end{aligned}
$$

Hence, proving the theorem.

## B   Limiting NNGP and NTK for `ReLU`

Consider the GP limit characterization of the pre-activation kernel $K^{(1)}(\mathbf{x}^{c,i},\mathbf{x}^{c',j})$ as follows:

$$
K_{GP}^{(1)}(\mathbf{x}^{c,i},\mathbf{x}^{c',j}) = \sigma_b^2 + \frac{\sigma_w^2}{d_0}\mathbf{x}^{c,i\top}\mathbf{x}^{c',j}. \tag{22}
$$

Observe that $K_{GP}^{(1)}(\mathbf{x}^{c,i},\mathbf{x}^{c',j})$ is independent of the activation function. Now, the closed form representation of the post-activation NNGP kernel $Q_{GP}^{(1)}(\cdot,\cdot)$ for the `ReLU` activation is given by:

$$
\begin{aligned}
Q_{GP-ReLU}^{(1)}(\mathbf{x}^{c,i},\mathbf{x}^{c',j}) &= \frac{\tau(x^{c,i},x^{c',j})}{2\pi}\sqrt{K_{GP}^{(1)}(\mathbf{x}^{c,i},\mathbf{x}^{c,i})K_{GP}^{(1)}(\mathbf{x}^{c',j},\mathbf{x}^{c',j})}, \\
\tau(x^{c,i},x^{c',j}) &= \sin\theta_{c,i}^{c',j} + \left(\pi - \theta_{c,i}^{c',j}\right)\cos\theta_{c,i}^{c',j} \\
\theta_{c,i}^{c',j} &= \arccos\left(\frac{K_{GP}^{(1)}(\mathbf{x}^{c,i},\mathbf{x}^{c',j})}{\sqrt{K_{GP}^{(1)}(\mathbf{x}^{c,i},\mathbf{x}^{c,i})K_{GP}^{(1)}(\mathbf{x}^{c',j},\mathbf{x}^{c',j})}}\right).
\end{aligned}
\tag{23}
$$

Next, we define the `ReLU` based derivative kernel $\dot{Q}_{GP-ReLU}^{(1)}(\cdot,\cdot)$ as follows:

$$
\dot{Q}_{GP-ReLU}^{(1)}(\mathbf{x}^{c,i},\mathbf{x}^{c',j}) = \frac{1}{2\pi}\left(\pi - \theta\right) \tag{24}
$$

Finally, the NTK can be formulated as follows:

$$
\Theta_{NTK-ReLU}^{(2)}(\mathbf{x}^{c,i},\mathbf{x}^{c',j}) = K_{GP-ReLU}^{(2)}(\mathbf{x}^{c,i},\mathbf{x}^{c',j}) + K_{GP}^{(1)}(\mathbf{x}^{c,i},\mathbf{x}^{c',j})\dot{Q}_{GP-ReLU}^{(1)}(\mathbf{x}^{c,i},\mathbf{x}^{c',j}). \tag{25}
$$

Here, $K_{GP-ReLU}^{(2)}(\mathbf{x}^{c,i},\mathbf{x}^{c',j})$ can be defined using the recursive formulation:

$$
K_{GP-ReLU}^{(2)}(\mathbf{x}^{c,i},\mathbf{x}^{c',j}) = \sigma_b^2 + \sigma_w^2 Q_{GP-ReLU}^{(1)}(\mathbf{x}^{c,i},\mathbf{x}^{c',j}). \tag{26}
$$

## C  General Results for NC1 with Kernels

In this section, we present some general results to calculate the expected value of $\mathbb{E}\left[\mathcal{NC}_1(\mathbf{H})\right]$ for any given kernel function $Q(\cdot,\cdot)$ that is associated with the features $\mathbf{H}$. To begin with, we consider a generic formulation of the three cases for $\mathbb{E}\left[Q(x^{c,i}, x^{c',j})\right]$:

$$\mathbb{E}\left[Q(x^{c,i}, x^{c',j})\right] = \begin{cases} V^{(1)}(c) & \text{if } c = c', i = j \\ V^{(2)}(c) & \text{if } c = c', i \neq j \\ V^{(3)}(c,c') & \text{if } c \neq c' \end{cases}. \tag{27}$$

**Lemma C.1.** *Given the cases for the expected values of a kernel function $Q(\cdot,\cdot)$ as per (27), the $\mathbb{E}\left[\operatorname{tr}(\boldsymbol{\Sigma}_W(\mathbf{H}))\right]$ is given by:*

$$\mathbb{E}\left[\operatorname{tr}(\boldsymbol{\Sigma}_W(\mathbf{H}))\right] = \sum_{c=1}^{2} \frac{n_c}{N} V^{(1)}(c) - \frac{1}{2n_c^2}\left(n_c(n_c-1)V^{(2)}(c) + n_c V^{(1)}(c)\right) \tag{28}$$

*Proof.* By leveraging Theorem 4.1, we can compute the expected value of $\operatorname{tr}(\boldsymbol{\Sigma}_W(\mathbf{H}))$ as follows:

$$\begin{aligned}
\mathbb{E}\left[\operatorname{tr}(\boldsymbol{\Sigma}_W(\mathbf{H}))\right] &= \mathbb{E}\left[\frac{1}{N}\sum_{c=1}^{C}\sum_{i=1}^{n_c} Q(x^{c,i}, x^{c,i})\right] - \mathbb{E}\left[\frac{1}{C}\sum_{c=1}^{C}\frac{1}{n_c^2}\sum_{i=1}^{n_c}\sum_{j=1}^{n_c} Q(x^{c,i}, x^{c,j})\right] \\
&= \frac{1}{N}\sum_{c=1}^{2}\sum_{i=1}^{n_c}\mathbb{E}\left[Q(x^{c,i}, x^{c,i})\right] - \frac{1}{2}\sum_{c=1}^{2}\frac{1}{n_c^2}\sum_{i=1}^{n_c}\sum_{j=1}^{n_c}\mathbb{E}\left[Q(x^{c,i}, x^{c,j})\right] \\
&= \frac{1}{N}\sum_{c=1}^{2}\sum_{i=1}^{n_c} V^{(1)}(c) - \frac{1}{2}\sum_{c=1}^{2}\frac{1}{n_c^2}\left(n_c(n_c-1)V^{(2)}(c) + n_c V^{(1)}(c)\right) \\
&= \sum_{c=1}^{2}\frac{n_c}{N}V^{(1)}(c) - \frac{1}{2n_c^2}\left(n_c(n_c-1)V^{(2)}(c) + n_c V^{(1)}(c)\right).
\end{aligned} \tag{29}$$

$\square$

**Lemma C.2.** *Given the cases for the expected values of a kernel function $Q(\cdot,\cdot)$ as per (27), the $\mathbb{E}\left[\operatorname{tr}(\boldsymbol{\Sigma}_B(\mathbf{H}))\right]$ is given by:*

$$\mathbb{E}\left[\operatorname{tr}(\boldsymbol{\Sigma}_B(\mathbf{H}))\right] = \left[\sum_{c=1}^{2}\left(\frac{1}{2n_c^2} - \frac{1}{N^2}\right)\left(n_c(n_c-1)V^{(2)}(c) + n_c V^{(1)}(c)\right)\right] - \frac{2n_1 n_2}{N^2}V^{(3)}(1,2) \tag{30}$$

*Proof.* The expected value of $\operatorname{tr}(\boldsymbol{\Sigma}_B(\mathbf{H}))$ can be computed using Theorem 4.1 as:

$$\begin{aligned}
\mathbb{E}\left[\operatorname{tr}(\boldsymbol{\Sigma}_B(\mathbf{H}))\right] &= \mathbb{E}\left[\frac{1}{C}\sum_{c=1}^{C}\frac{1}{n_c^2}\sum_{i=1}^{n_c}\sum_{j=1}^{n_c}\mathbf{Q}(x^{c,i}, x^{c,j})\right] - \mathbb{E}\left[\frac{1}{N^2}\sum_{c=1}^{C}\sum_{c'=1}^{C}\sum_{i=1}^{n_c}\sum_{j=1}^{n_{c'}}\mathbf{Q}(x^{c,i}, x^{c',j})\right] \\
&= \left[\frac{1}{2}\sum_{c=1}^{2}\frac{1}{n_c^2}\left(n_c(n_c-1)V^{(2)}(c) + n_c V^{(1)}(c)\right)\right] \\
&\quad - \frac{1}{N^2}\left[\sum_{c=1}^{2}\left(n_c(n_c-1)V^{(2)}(c) + n_c V^{(1)}(c)\right)\right] - \frac{1}{N^2}\left[2\sum_{i=1}^{n_1}\sum_{j=1}^{n_2}V^{(3)}(c=1, c'=2)\right] \\
&= \left[\sum_{c=1}^{2}\left(\frac{1}{2n_c^2} - \frac{1}{N^2}\right)\left(n_c(n_c-1)V^{(2)}(c) + n_c V^{(1)}(c)\right)\right] - \frac{2n_1 n_2}{N^2}V^{(3)}(1,2)
\end{aligned} \tag{31}$$

$\square$

**Lemma C.3.** *Given the cases for the expected values of a kernel function $Q(\cdot, \cdot)$ as per (27), the $\mathbb{E}[\mathcal{NC}_1(\mathbf{H})]$ is given by:*

$$\mathbb{E}[\mathcal{NC}_1(\mathbf{H})] = \frac{\sum\limits_{c=1}^{2} \frac{n_c V^{(1)}(c)}{N} - \frac{\left(n_c(n_c-1)V^{(2)}(c)+n_c V^{(1)}(c)\right)}{2n_c^2}}{\left[\sum\limits_{c=1}^{2}\left(\frac{1}{2n_c^2}-\frac{1}{N^2}\right)\left(n_c(n_c-1)V^{(2)}(c)+n_c V^{(1)}(c)\right)\right] - \frac{2n_1 n_2 V^{(3)}(1,2)}{N^2}} + \Delta_{h.o.t} \tag{32}$$

*Proof.* Note that the expectation of the ratios can be given as:

$$\mathbb{E}[\mathcal{NC}_1(\mathbf{H})] = \frac{\mathbb{E}[\text{tr}(\Sigma_W(\mathbf{H}))]}{\mathbb{E}[\text{tr}(\Sigma_B(\mathbf{H}))]} + \Delta_{h.o.t} \tag{33}$$

$$= \frac{\sum\limits_{c=1}^{2} \frac{n_c V^{(1)}(c)}{N} - \frac{\left(n_c(n_c-1)V^{(2)}(c)+n_c V^{(1)}(c)\right)}{2n_c^2}}{\left[\sum\limits_{c=1}^{2}\left(\frac{1}{2n_c^2}-\frac{1}{N^2}\right)\left(n_c(n_c-1)V^{(2)}(c)+n_c V^{(1)}(c)\right)\right] - \frac{2n_1 n_2 V^{(3)}(1,2)}{N^2}} + \Delta_{h.o.t} \tag{34}$$

Here, $\Delta_{h.o.t}$ corresponds to higher order terms given by (Seltman, 2012):

$$\Delta_{h.o.t} = \frac{Var(\text{tr}(\mathbf{\Sigma}_B(\mathbf{H})))\mathbb{E}[\text{tr}(\mathbf{\Sigma}_W(\mathbf{H}))]}{\mathbb{E}[\text{tr}(\Sigma_B(\mathbf{H}))]^3} - \frac{Cov(\text{tr}(\mathbf{\Sigma}_W(\mathbf{H})), \text{tr}(\mathbf{\Sigma}_B(\mathbf{H})))}{\mathbb{E}[\text{tr}(\Sigma_B(\mathbf{H}))]^2}, \tag{35}$$

where, based on the well-studied concentration of sample covariance matrices around the statistical covariance (Vershynin, 2012), $\Delta_{h.o.t}$ tend to 0 for large $n_c$ values.

$\square$

**Lemma C.4.** *For a random variable $x^{c,i} \sim \mathcal{N}(\mu_c, \sigma_c^2)$ which represents the $i^{th}$ sample of class $c$ (as per notation in Section 5.3), the expected value $\mathbb{E}\left[\frac{1}{(x^{c,i})^2}\right]$ is given by:*

$$T(c) = \mathbb{E}\left[\frac{1}{(x^{c,i})^2}\right] = \frac{1}{(\mu_c^2 + \sigma_c^2)} + \frac{2\sigma_c^4 + 4\sigma_c^2\mu_c^2}{(\mu_c^2 + \sigma_c^2)^3} \tag{36}$$

*Proof.* Based on the standard result on the expectation of ratios (Seltman, 2012), we get:

$$\mathbb{E}\left[\frac{1}{(x^{c,i})^2}\right] = \frac{1}{\mathbb{E}[(x^{c,i})^2]} + \frac{Var((x^{c,i})^2)}{\mathbb{E}[(x^{c,i})^2]^3} \tag{37}$$

$$= \frac{1}{(\mu_c^2 + \sigma_c^2)} + \frac{\mathbb{E}[(x^{c,i})^4] - (\mu_c^2 + \sigma_c^2)^2}{(\mu_c^2 + \sigma_c^2)^3} \tag{38}$$

Based on the results from the moment-generating function, we know that:

$$\mathbb{E}[(x^{c,i})^4] = 3\sigma_c^4 + 6\sigma_c^2\mu_c^2 + \mu_c^4, \tag{39}$$

which gives us:

$$\mathbb{E}\left[\frac{1}{(x^{c,i})^2}\right] = \frac{1}{(\mu_c^2 + \sigma_c^2)} + \frac{3\sigma_c^4 + 6\sigma_c^2\mu_c^2 + \mu_c^4 - (\mu_c^2 + \sigma_c^2)^2}{(\mu_c^2 + \sigma_c^2)^3} \tag{40}$$

$$= \frac{1}{(\mu_c^2 + \sigma_c^2)} + \frac{2\sigma_c^4 + 4\sigma_c^2\mu_c^2}{(\mu_c^2 + \sigma_c^2)^3}. \tag{41}$$

Hence proving the lemma.

$\square$

# D  Proof of Theorem 5.1

## D.1  NC1 of limiting NNGP with `ReLU` activation

In the limit $d_1 \to \infty$, we leverage the kernels in the GP limit as per (10), (23). Observe that for any two data points $x^{c,i}, x^{c',j} \in \mathbb{R}$, the value of $\theta_{c,i}^{c',j}$ can be given as:

$$\theta_{c,i}^{c',j} = \arccos \left( \frac{K_{GP}^{(1)}(x^{c,i}, x^{c',j})}{\sqrt{K_{GP}^{(1)}(x^{c,i}, x^{c,i}) K_{GP}^{(1)}(x^{c',j}, x^{c',j})}} \right)$$

$$= \arccos \left( \frac{\sigma_b^2 + \frac{\sigma_w^2}{d_0} x^{c,i} x^{c',j}}{\sqrt{\left( \sigma_b^2 + \frac{\sigma_w^2}{d_0} x^{c,i} x^{c,i} \right) \left( \sigma_b^2 + \frac{\sigma_w^2}{d_0} x^{c',j} x^{c',j} \right)}} \right).$$

Since $\sigma_b \to 0$, the value of $\theta_{c,i}^{c',j}$ simplifies to:

$$\theta_{c,i}^{c',j} = \begin{cases} 0 & \text{if } c = c' \\ \pi & \text{if } c \neq c' \end{cases}, \tag{42}$$

which follows from $\frac{x^{c,i} x^{c',j}}{\sqrt{x^{c,i} x^{c,i}} \sqrt{x^{c',j} x^{c',j}}} = \text{sign}(x^{c,i}) \, \text{sign}(x^{c',j})$ and $x^{1,i} < 0, x^{2,j} > 0$ almost surely. This leads to:

$$Q_{GP-ReLU}^{(1)}(x^{c,i}, x^{c',j}) = \frac{1}{2\pi} \sqrt{\sigma_w^4 (x^{c,i})^2 (x^{c',j})^2} \left( \sin \theta_{c,i}^{c',j} + \left( \pi - \theta_{c,i}^{c',j} \right) \cos \theta_{c,i}^{c',j} \right) \tag{43}$$

$$\implies Q_{GP-ReLU}^{(1)}(x^{c,i}, x^{c,j}) = \begin{cases} \frac{\sigma_w^2}{2} \left| x^{c,i} \right| \left| x^{c',j} \right| & \text{if } c = c' \\ 0 & \text{if } c \neq c' \end{cases} \tag{44}$$

For the $c = c'$ case, the value of the kernel boils down to the product of norms of independent random variables drawn from the same distribution. Since we assume $x^{c,i} x^{c',j} > 0$ if $c = c'$, the equation 44 can be rewritten as:

$$Q_{GP-ReLU}^{(1)}(x^{c,i}, x^{c,j}) = \begin{cases} \frac{\sigma_w^2}{2} x^{c,i} x^{c',j} & \text{if } c = c' \\ 0 & \text{if } c \neq c' \end{cases} \tag{45}$$

Additionally, since $x^{c,i}$ are random variables, the expected value of the kernel can be formulated as:

$$\mathbb{E}\left[ Q_{GP-ReLU}^{(1)}(x^{c,i}, x^{c',j}) \right] = \begin{cases} \frac{\sigma_w^2}{2} \left( \sigma_c^2 + \mu_c^2 \right) & \text{if } c = c', i = j \\ \frac{\sigma_w^2}{2} \mu_c^2 & \text{if } c = c', i \neq j \\ 0 & \text{if } c \neq c' \end{cases} \tag{46}$$

Thus, based on our generic formulation of cases in (27) in Appendix C, we get:

$$V^{(1)}(c) = \frac{\sigma_w^2}{2} \left( \sigma_c^2 + \mu_c^2 \right); \quad V^{(2)}(c) = \frac{\sigma_w^2}{2} \mu_c^2; \quad V^{(3)}(c, c') = 0. \tag{47}$$

As $N \gg 1$ and $n_c \gg 1, \forall c \in \{1, 2\}$, Lemma C.3 gives us:

$$\mathbb{E}[\mathcal{NC}_1(\mathbf{H}_{GP})] = \frac{\sum_{c=1}^2 \frac{n_c V^{(1)}(c)}{N} - \frac{V^{(2)}(c)}{2}}{\left[ \sum_{c=1}^2 \left( \frac{1}{2n_c^2} - \frac{1}{N^2} \right) \left( n_c^2 V^{(2)}(c) \right) \right] - \frac{2n_1 n_2}{N^2} V^{(3)}(1,2)} + \Delta_{h.o.t}$$

$$\implies \mathbb{E}[\mathcal{NC}_1(\mathbf{H}_{GP})] = \frac{\sum_{c=1}^2 \frac{n_c \mu_c^2 + n_c \sigma_c^2}{N} - \frac{\mu_c^2}{2}}{\left( \sum_{c=1}^2 \frac{\mu_c^2}{2} - \frac{n_c^2 \mu_c^2}{N^2} \right)} + \Delta_{h.o.t}. \tag{48}$$

## D.2 NC1 of limiting NTK with `ReLU` activation

The recursive relationship between the NTK and NNGP (Lee et al., 2019; Tirer et al., 2022) can be given as follows (13):

$$\Theta_{NTK-ReLU}^{(2)}(\mathbf{x}^{c,i}, \mathbf{x}^{c',j}) = K_{GP-ReLU}^{(2)}(\mathbf{x}^{c,i}, \mathbf{x}^{c',j}) + K_{GP}^{(1)}(\mathbf{x}^{c,i}, \mathbf{x}^{c',j})\dot{Q}_{GP-ReLU}^{(1)}(\mathbf{x}^{c,i}, \mathbf{x}^{c',j}) \tag{49}$$

Here, $K_{GP-ReLU}^{(2)}(\mathbf{x}^{c,i}, \mathbf{x}^{c',j})$ can be defined using the following recursive formulation:

$$K_{GP-ReLU}^{(2)}(\mathbf{x}^{c,i}, \mathbf{x}^{c',j}) = \sigma_b^2 + \sigma_w^2 Q_{GP-ReLU}^{(1)}(\mathbf{x}^{c,i}, \mathbf{x}^{c',j}). \tag{50}$$

Based on (24), the derivative $\dot{Q}_{GP-ReLU}^{(1)}$ can be given as follows:

$$\dot{Q}_{GP-ReLU}^{(1)}(\mathbf{x}^{c,i}, \mathbf{x}^{c',j}) = \frac{1}{2\pi}\left(\pi - \theta_{c,i}^{c',j}\right)$$

$$\theta_{c,i}^{c',j} = \arccos\left(\frac{K_{GP}^{(1)}(\mathbf{x}^{c,i}, \mathbf{x}^{c',j})}{\sqrt{K_{GP}^{(1)}(\mathbf{x}^{c,i}, \mathbf{x}^{c,i})K_{GP}^{(1)}(\mathbf{x}^{c',j}, \mathbf{x}^{c',j})}}\right). \tag{51}$$

We build on the results from the NNGP analysis (with $Q_{GP-ReLU}^{(1)}(\mathbf{x}^{c,i}, \mathbf{x}^{c',j})$) for computing the variability collapse with the limiting NTK. First, note that:

$$\theta_{c,i}^{c',j} = \begin{cases} 0 & \text{if } c = c' \\ \pi & \text{if } c \neq c' \end{cases}. \tag{52}$$

When $\sigma_b \to 0$, we get:

$$\Theta_{NTK-ReLU}^{(2)}(\mathbf{x}^{c,i}, \mathbf{x}^{c',j}) = \sigma_w^2 Q_{GP-ReLU}^{(1)}(\mathbf{x}^{c,i}, \mathbf{x}^{c',j}) + K_{GP}^{(1)}(\mathbf{x}^{c,i}, \mathbf{x}^{c',j})\dot{Q}_{GP-ReLU}^{(1)}(\mathbf{x}^{c,i}, \mathbf{x}^{c',j}). \tag{53}$$

From (45), we know that:

$$Q_{GP-ReLU}^{(1)}(x^{c,i}, x^{c,j}) = \begin{cases} \frac{\sigma_w^2}{2}x^{c,i}x^{c',j} & \text{if } c = c' \\ 0 & \text{if } c \neq c' \end{cases} \tag{54}$$

$$\implies \Theta_{NTK-ReLU}^{(2)}(x^{c,i}, x^{c',j}) = \begin{cases} \frac{\sigma_w^4}{2}x^{c,i}x^{c,j} + \frac{\sigma_w^2}{2}x^{c,i}x^{c,j} & \text{if } c = c' \\ 0 & \text{if } c \neq c' \end{cases}, \tag{55}$$

$$= \begin{cases} \left(\frac{\sigma_w^4}{2} + \frac{\sigma_w^2}{2}\right)x^{c,i}x^{c,j} & \text{if } c = c' \\ 0 & \text{if } c \neq c' \end{cases}. \tag{56}$$

Notice that $\Theta_{NTK-ReLU}^{(2)}(x^{c,i}, x^{c',j})$ is a scaled version of $Q_{GP-ReLU}^{(1)}(x^{c,i}, x^{c',j})$ (as per (45)). Thus, we end up with the same result as (48) :

$$\mathbb{E}\left[\mathcal{NC}_1(\mathbf{H}_{NTK})\right] = \frac{\sum_{c=1}^2 \frac{n_c\mu_c^2 + n_c\sigma_c^2}{N} - \frac{\mu_c^2}{2}}{\left(\sum_{c=1}^2 \frac{\mu_c^2}{2} - \frac{n_c^2\mu_c^2}{N^2}\right)} + \Delta_{h.o.t.} \tag{57}$$

# E    Results for NC1 with `Erf` activation

## E.1    NC1 of Limiting NNGP with `Erf` activation

Under the Assumptions described in Section 5.3 with $d_0 = 1, d_1 \to \infty$, observe that (11) gives us:

$$Q^{(1)}_{GP-Erf}(x^{c,i}, x^{c',j}) = \frac{2}{\pi} \arcsin \left( \frac{2K^{(1)}_{GP}(x^{c,i}, x^{c',j})}{\sqrt{1 + 2K^{(1)}_{GP}(x^{c,i}, x^{c,i})}\sqrt{1 + 2K^{(1)}_{GP}(x^{c',j}, x^{c',j})}} \right) \tag{58}$$

$$= \frac{2}{\pi} \arcsin \left( \frac{2\sigma_b^2 + 2\sigma_w^2 x^{c,i} x^{c',j}}{\sqrt{1 + 2\sigma_b^2 + 2\sigma_w^2 (x^{c,i})^2}\sqrt{1 + 2\sigma_b^2 + 2\sigma_w^2 (x^{c',j})^2}} \right) \tag{59}$$

Considering $\sigma_b \to 0$:

$$Q^{(1)}_{GP-Erf}(x^{c,i}, x^{c',j}) = \frac{2}{\pi} \arcsin \left( \frac{2\sigma_w^2 x^{c,i} x^{c',j}}{\sqrt{1 + 2\sigma_w^2 (x^{c,i})^2}\sqrt{1 + 2\sigma_w^2 (x^{c',j})^2}} \right) \tag{60}$$

$$= \frac{2}{\pi} \arcsin \left( \frac{\text{sign}(x^{c,i})\,\text{sign}(x^{c',j})}{\sqrt{1 + \frac{1}{2\sigma_w^2 (x^{c,i})^2}}\sqrt{1 + \frac{1}{2\sigma_w^2 (x^{c',j})^2}}} \right), \tag{61}$$

where the last equality comes from:

$$\frac{x^{c,i} x^{c',j}}{|x^{c,i}|\sqrt{1 + \frac{1}{2\sigma_w^2 (x^{c,i})^2}} \cdot |x^{c',j}|\sqrt{1 + \frac{1}{2\sigma_w^2 (x^{c',j})^2}}} = \frac{\text{sign}(x^{c,i})\,\text{sign}(x^{c',j})}{\sqrt{1 + \frac{1}{2\sigma_w^2 (x^{c,i})^2}}\sqrt{1 + \frac{1}{2\sigma_w^2 (x^{c',j})^2}}}. \tag{62}$$

For notational simplicity, consider:

$$\rho(x^{c,i}, x^{c',j}) = \sqrt{1 + \frac{1}{2\sigma_w^2 (x^{c,i})^2}}\sqrt{1 + \frac{1}{2\sigma_w^2 (x^{c',j})^2}}, \tag{63}$$

and represent $Q^{(1)}_{GP-Erf}(x^{c,i}, x^{c',j})$ as:

$$Q^{(1)}_{GP-Erf}(x^{c,i}, x^{c',j}) = \frac{2}{\pi} \arcsin \left( \frac{\text{sign}(x^{c,i})\,\text{sign}(x^{c',j})}{\rho(x^{c,i}, x^{c',j})} \right). \tag{64}$$

Based on Assumption 1, we know that $x^{1,i} < 0, x^{2,j} > 0$ almost surely. This leads to:

$$Q^{(1)}_{GP-Erf}(x^{c,i}, x^{c',j}) = \begin{cases} \frac{2}{\pi} \arcsin \left( \frac{1}{\rho(x^{c,i}, x^{c,j})} \right) & \text{if } c = c' \\ -\frac{2}{\pi} \arcsin \left( \frac{1}{\rho(x^{c,i}, x^{c',j})} \right) & \text{if } c \neq c' \end{cases}. \tag{65}$$

### E.1.1    Calculating $\mathbb{E}\left[ Q^{(1)}_{GP-Erf}(x^{c,i}, x^{c',j}) \right]$

For $|u| \le 1$, we consider the expansion of $\arcsin(u) = u + \frac{u^3}{6} + \cdots$ to obtain:

$$\mathbb{E}\left[ \arcsin \left( \frac{1}{\rho(x^{c,i}, x^{c',j})} \right) \right] = \mathbb{E}\left[ \frac{1}{\rho(x^{c,i}, x^{c',j})} \right] + \mathbb{E}\left[ \frac{1}{6\rho(x^{c,i}, x^{c',j})^3} \right] + \cdots. \tag{66}$$

To this end, based on Assumption 1 of large enough $|\mu_1|, |\mu_2|$, we approximate the expectation with only the first term and denote $\xi_{h.o.t}$ to capture the effects of the higher order terms. Notice that since $\rho(x^{c,i}, x^{c',j}) > 1$

for finite $(x^{c,i}, x^{c',j})$, the effects of $\xi_{h.o.t}$ are finite but decay rapidly compared to the first term. To this end, we get:

$$\mathbb{E}\left[\arcsin\left(\frac{1}{\rho(x^{c,i}, x^{c',j})}\right)\right] = \mathbb{E}\left[\frac{1}{\rho(x^{c,i}, x^{c',j})}\right] + \xi_{h.o.t} \tag{67}$$

Calculating the expectation $\mathbb{E}\left[\frac{1}{\rho(x^{c,i}, x^{c',j})}\right]$ can now be split based on $c, c'$.

• Case $c = c', i = j$:

$$\rho(x^{c,i}, x^{c,i}) = 1 + \frac{1}{2\sigma_w^2(x^{c,i})^2} \tag{68}$$

$$\implies \mathbb{E}\left[\rho(x^{c,i}, x^{c,i})\right] = 1 + \frac{1}{2\sigma_w^2}\mathbb{E}\left[\frac{1}{(x^{c,i})^2}\right] \tag{69}$$

$$= 1 + \frac{T(c)}{2\sigma_w^2}. \tag{70}$$

The last equality is based on Lemma C.4 which gives the expanded version of $T(c)$.

Finally, the value of $\mathbb{E}\left[\frac{1}{\rho(x^{c,i}, x^{c,i})}\right]$ can be given as:

$$\mathbb{E}\left[\frac{1}{\rho(x^{c,i}, x^{c,i})}\right] = \frac{1}{\mathbb{E}\left[\rho(x^{c,i}, x^{c,i})\right]} + \frac{Var(\rho(x^{c,i}, x^{c,i}))}{\mathbb{E}\left[\rho(x^{c,i}, x^{c,i})\right]^3} \tag{71}$$

$$= \frac{1}{1 + \frac{T(c)}{2\sigma_w^2}} + \delta_{h.o.t}(\rho(x^{c,i}, x^{c,i})) \tag{72}$$

Notice that even in this simple case, the expressions are non-trivial to fully expand. Nonetheless, along with Assumption 1, we consider large enough $|\mu_1|, |\mu_2|$ such that:

$$\frac{T(c)}{2\sigma_w^2} = \frac{1}{2\sigma_w^2}\left[\frac{1}{(\mu_c^2 + \sigma_c^2)} + \frac{2\sigma_c^4 + 4\sigma_c^2\mu_c^2}{(\mu_c^2 + \sigma_c^2)^3}\right] < 1. \tag{73}$$

Thus, based on the expansion of $(1+u)^{-1} = 1 - u + u^2 - u^3 + \cdots$, we obtain the following cleaner approximation:

$$\mathbb{E}\left[\frac{1}{\rho(x^{c,i}, x^{c,i})}\right] = 1 - \frac{T(c)}{2\sigma_w^2} + \Delta_{h.o.t}^{(1)}(c). \tag{74}$$

Here $\Delta_{h.o.t}^{(1)}(c)$ captures all the higher order terms corresponding to $\left(\frac{T(c)}{2\sigma_w^2}\right)^2 - \left(\frac{T(c)}{2\sigma_w^2}\right)^3 + \cdots$ and $\delta_{h.o.t}(\rho(x^{c,i}, x^{c,i}))$ as denoted above.

• Case $c = c', i \neq j$:

In the case of $c = c', i \neq j$, the expectations on the square roots do not have a particular closed form. To this, end we leverage Assumption 1 to obtain the following approximation:

$$\rho(x^{c,i}, x^{c,j}) = \sqrt{1 + \frac{1}{2\sigma_w^2(x^{c,i})^2}}\sqrt{1 + \frac{1}{2\sigma_w^2(x^{c,j})^2}} \tag{75}$$

$$= \left(1 + \frac{1}{4\sigma_w^2(x^{c,i})^2} + h.o.t\right)\left(1 + \frac{1}{4\sigma_w^2(x^{c,j})^2} + h.o.t\right) \tag{76}$$

$$\implies \mathbb{E}\left[\rho(x^{c,i}, x^{c,j})\right] = \mathbb{E}\left[1 + \frac{1}{4\sigma_w^2(x^{c,i})^2} + h.o.t\right]\mathbb{E}\left[1 + \frac{1}{4\sigma_w^2(x^{c,j})^2} + h.o.t\right] \tag{77}$$

Observe that the inner terms in the expectations are scaled versions of the above case. To this end, we approximate $\mathbb{E}\left[\frac{1}{\rho(x^{c,i}, x^{c,j})}\right]$ as:

$$\mathbb{E}\left[\frac{1}{\rho(x^{c,i}, x^{c,j})}\right] \approx \frac{1}{\left(1 + \frac{T(c)}{4\sigma_w^2}\right)^2} + \delta_{h.o.t}(\rho(x^{c,i}, x^{c,j})) \tag{78}$$

$$= \frac{1}{1 + \frac{T(c)}{2\sigma_w^2} + \frac{T(c)^2}{16\sigma_w^4}} + \delta_{h.o.t}(\rho(x^{c,i}, x^{c,j})) \tag{79}$$

Similar to the assumption that led to (74), we get:

$$\mathbb{E}\left[\frac{1}{\rho(x^{c,i}, x^{c,j})}\right] \approx 1 - \frac{T(c)}{2\sigma_w^2} - \frac{T(c)^2}{16\sigma_w^4} + \Delta_{h.o.t}^{(2)}(c). \tag{80}$$

- **Case** $c \neq c'$

A similar analysis as above applies in this case:

$$\rho(x^{c,i}, x^{c',j}) = \sqrt{1 + \frac{1}{2\sigma_w^2(x^{c,i})^2}} \sqrt{1 + \frac{1}{2\sigma_w^2(x^{c',j})^2}} \tag{81}$$

$$= \left(1 + \frac{1}{4\sigma_w^2(x^{c,i})^2} + h.o.t\right)\left(1 + \frac{1}{4\sigma_w^2(x^{c',j})^2} + h.o.t\right) \tag{82}$$

$$\implies \mathbb{E}\left[\rho(x^{c,i}, x^{c',j})\right] = \mathbb{E}\left[1 + \frac{1}{4\sigma_w^2(x^{c,i})^2} + h.o.t\right]\mathbb{E}\left[1 + \frac{1}{4\sigma_w^2(x^{c',j})^2} + h.o.t\right] \tag{83}$$

Observe that the inner terms in the expectations are similar to the above case. To this end, we approximate $\mathbb{E}\left[\frac{1}{\rho(x^{c,i}, x^{c',j})}\right]$ as:

$$\mathbb{E}\left[\frac{1}{\rho(x^{c,i}, x^{c',j})}\right] \approx \frac{1}{\left(1 + \frac{T(c)}{4\sigma_w^2}\right)\left(1 + \frac{T(c')}{4\sigma_w^2}\right)} + \delta_{h.o.t}(\rho(x^{c,i}, x^{c',j})) \tag{84}$$

$$= \frac{1}{1 + \frac{T(c)+T(c')}{4\sigma_w^2} + \frac{T(c)T(c')}{16\sigma_w^4}} + \delta_{h.o.t}(\rho(x^{c,i}, x^{c',j})) \tag{85}$$

Similar to the assumption that led to (74), we get:

$$\mathbb{E}\left[\frac{1}{\rho(x^{c,i}, x^{c',j})}\right] \approx 1 - \frac{T(c) + T(c')}{4\sigma_w^2} - \frac{T(c)T(c')}{16\sigma_w^4} + \Delta_{h.o.t}^{(3)}(c, c'). \tag{86}$$

Finally, based on (74), (80), (86) we obtain the following result for $\mathbb{E}[Q_{GP-Erf}^{(1)}(x^{c,i}, x^{c',j})]$ as :

$$\mathbb{E}\left[Q_{GP-Erf}^{(1)}(x^{c,i}, x^{c',j})\right]$$
$$\approx \begin{cases} 1 - \frac{T(c)}{2\sigma_w^2} + \Delta_{h.o.t}^{(1)}(c) & \text{if } c = c', i = j \\ 1 - \frac{T(c)}{2\sigma_w^2} - \frac{T(c)^2}{16\sigma_w^4} + \Delta_{h.o.t}^{(2)}(c) & \text{if } c = c', i \neq j \\ 1 - \frac{T(c)+T(c')}{4\sigma_w^2} - \frac{T(c)T(c')}{16\sigma_w^4} + \Delta_{h.o.t}^{(3)}(c, c') & \text{if } c \neq c' \end{cases} \tag{87}$$

Here $\Delta_{h.o.t}^{(1)}(c), \Delta_{h.o.t}^{(2)}(c), \Delta_{h.o.t}^{(3)}(c, c')$ are the collective higher order terms that tend to 0 as $|\mu_c|$ increases relative to smaller values of $\sigma_c$. These cases can now be plugged into our generic formulation of expected

values of a kernel function (i.e $V^{(1)}(c), V^{(2)}(c), V^{(3)}(c, c')$) as per (27) in Appendix C. Thus, based on Lemma C.3 for sufficiently large $\{n_c\}$ we get :

$$\mathbb{E}[\mathcal{NC}_1(\mathbf{H})] = \frac{\sum_{c=1}^{2} \frac{n_c V^{(1)}(c)}{N} - \frac{V^{(2)}(c)}{2}}{\left[\sum_{c=1}^{2} \left(\frac{1}{2n_c^2} - \frac{1}{N^2}\right)\left(n_c^2 V^{(2)}(c)\right)\right] - \frac{2n_1 n_2}{N^2} V^{(3)}(1, 2)} + \Delta_{h.o.t} \tag{88}$$

• **Numerator in the balanced class setting.**

To better understand the result, let's consider the balanced class scenario with $n_1 = n_2 = N/2$, for which the numerator simplifies to:

$$\sum_{c=1}^{2} \frac{n_c V^{(1)}(c)}{N} - \frac{V^{(2)}(c)}{2} = \sum_{c=1}^{2} \frac{V^{(1)}(c) - V^{(2)}(c)}{2} \tag{89}$$

$$= \sum_{c=1}^{2} \frac{\frac{T(c)^2}{16\sigma_w^4} + \Delta_{h.o.t}^{(1)}(c) - \Delta_{h.o.t}^{(2)}(c)}{2}. \tag{90}$$

If we were to ignore the effects of the higher order terms, then observe that the numerator primarily depends on $T(c)^2$, which can be given based on Lemma C.4 as:

$$T(c)^2 = \left[\frac{1}{(\mu_c^2 + \sigma_c^2)} + \frac{2\sigma_c^4 + 4\sigma_c^2 \mu_c^2}{(\mu_c^2 + \sigma_c^2)^3}\right]^2 \tag{91}$$

Thus, showcasing the dependence on $\mu_c, \sigma_c$ in determining the extent of collapse. For sufficiently large $|\mu_c| \gg \sigma_c$, we can approximate this value to:

$$T(c)^2 \approx \left[\frac{1}{\mu_c^2} + \frac{4\sigma_c^2}{\mu_c^4}\right]^2 = \frac{1}{\mu_c^4}\left[1 + \frac{4\sigma_c^2}{\mu_c^2}\right]^2 = \frac{1}{\mu_c^4}\left[1 + \frac{8\sigma_c^2}{\mu_c^2} + \frac{16\sigma_c^4}{\mu_c^4}\right] \tag{92}$$

• **Denominator in the balanced class setting.**

Similar to the numerator analysis, observe that when $n_1 = n_2 = N/2$, the denominator can be given as:

$$\left[\sum_{c=1}^{2}\left(\frac{1}{2n_c^2} - \frac{1}{N^2}\right)\left(n_c^2 V^{(2)}(c)\right)\right] - \frac{2n_1 n_2}{N^2} V^{(3)}(1, 2) \tag{93}$$

$$= \frac{V^{(2)}(1) + V^{(2)}(2) - 2V^{(3)}(1, 2)}{4} \tag{94}$$

$$= \frac{-\frac{T(1)}{2\sigma_w^2} - \frac{T(1)^2}{16\sigma_w^4} + \Delta_{h.o.t}^{(2)}(1) - \frac{T(2)}{2\sigma_w^2} - \frac{T(2)^2}{16\sigma_w^4} + \Delta_{h.o.t}^{(2)}(2)}{4} \tag{95}$$

$$+ \frac{2\frac{T(1)+T(2)}{4\sigma_w^2} + 2\frac{T(1)T(2)}{16\sigma_w^4} - 2\Delta_{h.o.t}^{(3)}(1, 2)}{4} \tag{96}$$

$$= \frac{-\frac{T(1)^2}{16\sigma_w^4} - \frac{T(2)^2}{16\sigma_w^4} + 2\frac{T(1)T(2)}{16\sigma_w^4} + \Delta_{h.o.t}^{(2)}(1) + \Delta_{h.o.t}^{(2)}(2) - 2\Delta_{h.o.t}^{(3)}(1, 2)}{4} \tag{97}$$

$$= \frac{-\left(\frac{T(1)-T(2)}{4\sigma_w^2}\right)^2 + \Delta_{h.o.t}^{(2)}(1) + \Delta_{h.o.t}^{(2)}(2) - 2\Delta_{h.o.t}^{(3)}(1, 2)}{4}. \tag{98}$$

Observe that the term $T(1) - T(2)$ represents:

$$T(1) - T(2) = \left[\frac{1}{(\mu_1^2 + \sigma_1^2)} + \frac{2\sigma_1^4 + 4\sigma_1^2 \mu_1^2}{(\mu_1^2 + \sigma_1^2)^3}\right] - \left[\frac{1}{(\mu_2^2 + \sigma_2^2)} + \frac{2\sigma_2^4 + 4\sigma_2^2 \mu_2^2}{(\mu_2^2 + \sigma_2^2)^3}\right] \tag{99}$$

and for sufficiently large $|\mu_c| \gg \sigma_c$, essentially represents:

$$T(1) - T(2) \approx \frac{1}{\mu_1^2} + \frac{4\sigma_1^2}{\mu_1^4} - \frac{1}{\mu_2^2} - \frac{4\sigma_2^2}{\mu_2^4}. \tag{100}$$

### E.2 NC1 of Limiting NTK with `Erf` activation

Recall from (13) that the recursive relationship between the NTK and NNGP can be given as follows:

$$\Theta_{NTK-Erf}^{(2)}(x^{c,i}, x^{c',j}) = K_{GP-Erf}^{(2)}(x^{c,i}, x^{c',j}) + K_{GP}^{(1)}(x^{c,i}, x^{c',j})\dot{Q}_{GP-Erf}^{(1)}(x^{c,i}, x^{c',j}), \tag{101}$$

where:

$$K_{GP-Erf}^{(2)}(x^{c,i}, x^{c',j}) = \sigma_b^2 + \sigma_w^2 Q_{GP-Erf}^{(1)}(x^{c,i}, x^{c',j}) \tag{102}$$

$$Q_{GP-Erf}^{(1)}(x^{c,i}, x^{c',j}) = \frac{2}{\pi} \arcsin\left(\frac{2K_{GP}^{(1)}(x^{c,i}, x^{c',j})}{\sqrt{1 + 2K_{GP}^{(1)}(x^{c,i}, x^{c,i})}\sqrt{1 + 2K_{GP}^{(1)}(x^{c',j}, x^{c',j})}}\right) \tag{103}$$

$$\dot{Q}_{GP-Erf}^{(1)}(x^{c,i}, x^{c',j}) = \frac{4}{\pi}\left[\left(1 + 2K_{GP}^{(1)}(x^{c,i}, x^{c,i})\right)\left(1 + 2K_{GP}^{(1)}(x^{c',j}, x^{c',j})\right) - \right.$$
$$\left.\left(2K_{GP}^{(1)}(x^{c,i}, x^{c',j})\right)^2\right]^{-1/2} \tag{104}$$

Considering $\sigma_b \to 0$, $d_0 = 1$ (as per the setting and assumptions), we get:

$$K_{GP}^{(1)}(\mathbf{x}^{c,i}, \mathbf{x}^{c',j}) = \sigma_w^2 x^{c,i} x^{c',j} \tag{105}$$

$$Q_{GP-Erf}^{(1)}(x^{c,i}, x^{c',j}) = \frac{2}{\pi} \arcsin\left(\frac{2\sigma_w^2 x^{c,i} x^{c',j}}{\sqrt{1 + 2\sigma_w^2 (x^{c,i})^2}\sqrt{1 + 2\sigma_w^2 (x^{c',j})^2}}\right). \tag{106}$$

$$\dot{Q}_{GP-Erf}^{(1)}(x^{c,i}, x^{c',j}) = \frac{4}{\pi}\left(\left(1 + 2\sigma_w^2 x^{c,i} x^{c,i}\right)\left(1 + 2\sigma_w^2 x^{c',j} x^{c',j}\right) - \left(2\sigma_w^2 x^{c,i} x^{c',j}\right)^2\right)^{-1/2}$$
$$= \frac{4}{\pi\sqrt{1 + 2\sigma_w^2 \cdot (x^{c,i})^2 + 2\sigma_w^2 \cdot (x^{c',j})^2}}. \tag{107}$$

This gives us:

$$K_{GP}^{(1)}(\mathbf{x}^{c,i}, \mathbf{x}^{c',j})\dot{Q}_{GP-Erf}^{(1)}(x^{c,i}, x^{c',j}) = \frac{4\sigma_w^2 x^{c,i} x^{c',j}}{\pi\sqrt{1 + 2\sigma_w^2 \cdot (x^{c,i})^2 + 2\sigma_w^2 \cdot (x^{c',j})^2}} \tag{108}$$

$$= \frac{4\sigma_w^2 x^{c,i} x^{c',j}}{\pi\sigma_w |x^{c,i}||x^{c',j}|\sqrt{\frac{1}{\sigma_w^2 (x^{c,i})^2 (x^{c',j})^2} + \frac{2}{(x^{c',j})^2} + \frac{2}{(x^{c,i})^2}}} \tag{109}$$

$$= \frac{4\sigma_w \operatorname{sign}(x^{c,i})\operatorname{sign}(x^{c',j})}{\pi\sqrt{\frac{1}{\sigma_w^2 (x^{c,i})^2 (x^{c',j})^2} + \frac{2}{(x^{c',j})^2} + \frac{2}{(x^{c,i})^2}}} \tag{110}$$

For notational simplicity, consider:

$$\kappa(x^{c,i}, x^{c',j}) = \sqrt{\frac{1}{\sigma_w^2 (x^{c,i})^2 (x^{c',j})^2} + \frac{2}{(x^{c',j})^2} + \frac{2}{(x^{c,i})^2}} \tag{111}$$

which simplifies the kernel formulation to:

$$\Theta_{NTK-Erf}^{(2)}(x^{c,i}, x^{c',j}) = \sigma_w^2 Q_{GP-Erf}^{(1)}(x^{c,i}, x^{c',j}) + \frac{4\sigma_w \operatorname{sign}(x^{c,i}) \operatorname{sign}(x^{c',j})}{\pi \kappa(x^{c,i}, x^{c',j})} \tag{112}$$

### E.2.1  Calculating $\mathbb{E}\left[\Theta_{NTK-Erf}^{(2)}(x^{c,i}, x^{c',j})\right]$

Similar to the NNGP analysis, we break down the calculation of $\mathbb{E}\left[\kappa(x^{c,i}, x^{c',j})\right]$ into three cases.

- **Case** $c = c', i = j$

$$\kappa(x^{c,i}, x^{c,i}) = \sqrt{\frac{1}{\sigma_w^2 (x^{c,i})^4} + \frac{4}{(x^{c,i})^2}} = \sqrt{1 - \left(1 - \frac{1}{\sigma_w^2 (x^{c,i})^4} - \frac{4}{(x^{c,i})^2}\right)} \tag{113}$$

$$= 1 - \frac{1}{2}\left(1 - \frac{1}{\sigma_w^2 (x^{c,i})^4} - \frac{4}{(x^{c,i})^2}\right) + \xi_{h.o.t} \tag{114}$$

$$= \frac{1}{2} + \frac{1}{2\sigma_w^2 (x^{c,i})^4} + \frac{2}{(x^{c,i})^2} + \xi_{h.o.t} \tag{115}$$

This gives us:

$$
\begin{aligned}
\mathbb{E}\left[\kappa(x^{c,i}, x^{c,i})\right] &= \frac{1}{2} + \mathbb{E}\left[\frac{1}{2\sigma_w^2 (x^{c,i})^4}\right] + \mathbb{E}\left[\frac{2}{(x^{c,i})^2}\right] + \mathbb{E}[\xi_{h.o.t}] \\
&= \frac{1}{2} + \frac{1}{2\sigma_w^2}\left[\frac{1}{\mathbb{E}\left[(x^{c,i})^4\right]} + \frac{Var((x^{c,i})^4)}{\mathbb{E}\left[(x^{c,i})^4\right]^3}\right] + 2\left[\frac{1}{\mathbb{E}\left[(x^{c,i})^2\right]} + \frac{Var((x^{c,i})^2)}{\mathbb{E}\left[(x^{c,i})^2\right]^3}\right] \\
&\quad + \mathbb{E}[\xi_{h.o.t}]
\end{aligned}
\tag{116}
$$

Based on the results from the moment-generating function, we know that:

$$\mathbb{E}[(x^{c,i})^4] = 3\sigma_c^4 + 6\sigma_c^2 \mu_c^2 + \mu_c^4, \tag{117}$$

which can be used along with Lemma C.4 to obtain:

$$\mathbb{E}\left[\kappa(x^{c,i}, x^{c,i})\right] = \frac{1}{2} + 2T(c) + \mathbb{E}[\xi_{h.o.t}] \tag{118}$$

For notational simplicity, we define a helper function as follows:

$$S(\mu_c, \sigma_c) = -\frac{1}{2} + 2T(c) + \mathbb{E}[\xi_{h.o.t}], \tag{119}$$

which gives us:

$$\mathbb{E}\left[\kappa(x^{c,i}, x^{c,i})\right] = 1 + S(\mu_c, \sigma_c) \tag{120}$$

Finally, the value of $\mathbb{E}\left[\frac{1}{\kappa(x^{c,i}, x^{c,i})}\right]$ can be given as:

$$\mathbb{E}\left[\frac{1}{\kappa(x^{c,i}, x^{c,i})}\right] = \frac{1}{\mathbb{E}\left[\kappa(x^{c,i}, x^{c,i})\right]} + \frac{Var(\kappa(x^{c,i}, x^{c,i}))}{\mathbb{E}\left[\kappa(x^{c,i}, x^{c,i})\right]^3} \tag{121}$$

$$= \frac{1}{1 + S(\mu_c, \sigma_c)} + \delta_{h.o.t}(\kappa(x^{c,i}, x^{c,i})) \tag{122}$$

Notice that even in this simple case, the expressions are non-trivial to fully expand. Nonetheless, along with Assumption 1, we consider large enough $|\mu_1|, |\mu_2|$ such that:

$$S(\mu_c, \sigma_c) < 1. \tag{123}$$

Thus, based on the expansion of $(1+u)^{-1} = 1 - u + u^2 - u^3 + \cdots$, we obtain the following cleaner approximation:

$$\mathbb{E}\left[\frac{1}{\kappa(x^{c,i}, x^{c,i})}\right] = 1 - S(\mu_c, \sigma_c) + \widetilde{\delta}_{h.o.t}(\kappa(x^{c,i}, x^{c,i})) \tag{124}$$

- **Case** $c = c', i \neq j$:

$$\kappa(x^{c,i}, x^{c,j}) = \sqrt{\frac{1}{\sigma_w^2 (x^{c,i})^2 (x^{c,j})^2} + \frac{2}{(x^{c,j})^2} + \frac{2}{(x^{c,i})^2}} \tag{125}$$

$$= \sqrt{1 - \left(1 - \frac{1}{\sigma_w^2 (x^{c,i})^2 (x^{c,j})^2} - \frac{2}{(x^{c,j})^2} - \frac{2}{(x^{c,i})^2}\right)} \tag{126}$$

$$= 1 - \frac{1}{2}\left(1 - \frac{1}{\sigma_w^2 (x^{c,i})^2 (x^{c,j})^2} - \frac{2}{(x^{c,j})^2} - \frac{2}{(x^{c,i})^2}\right) + \xi'_{h.o.t} \tag{127}$$

$$= \frac{1}{2} + \frac{1}{2\sigma_w^2 (x^{c,i})^2 (x^{c,j})^2} + \frac{1}{(x^{c,j})^2} + \frac{1}{(x^{c,i})^2} + \xi'_{h.o.t} \tag{128}$$

Thus, based on Lemma C.4, we get:

$$\mathbb{E}\left[\kappa(x^{c,i}, x^{c,j})\right] = \frac{1}{2} + \mathbb{E}\left[\frac{1}{2\sigma_w^2 (x^{c,i})^2 (x^{c,j})^2}\right] + \mathbb{E}\left[\frac{1}{(x^{c,j})^2}\right] + \mathbb{E}\left[\frac{1}{(x^{c,i})^2}\right] + \mathbb{E}[\xi'_{h.o.t}] \tag{129}$$

$$= \frac{1}{2} + \frac{1}{2\sigma_w^2}\mathbb{E}\left[\frac{1}{(x^{c,i})^2}\right]\mathbb{E}\left[\frac{1}{(x^{c,j})^2}\right] + \mathbb{E}\left[\frac{1}{(x^{c,j})^2}\right] + \mathbb{E}\left[\frac{1}{(x^{c,i})^2}\right] + \mathbb{E}[\xi'_{h.o.t}] \tag{130}$$

$$= \frac{1}{2} + \frac{T(c)^2}{2\sigma_w^2} + 2T(c) + \mathbb{E}[\xi'_{h.o.t}] \tag{131}$$

This leads to:

$$\mathbb{E}\left[\frac{1}{\kappa(x^{c,i}, x^{c,j})}\right] = \mathbb{E}\left[\frac{1}{1 + \left(-\frac{1}{2} + \frac{T(c)^2}{2\sigma_w^2} + 2T(c) + \mathbb{E}[\xi'_{h.o.t}]\right)}\right] \tag{132}$$

$$= 1 - \left(-\frac{1}{2} + \frac{T(c)^2}{2\sigma_w^2} + 2T(c) + \mathbb{E}[\xi'_{h.o.t}]\right) + \delta'_{h.o.t}(\kappa(x^{c,i}, x^{c,j})) \tag{133}$$

$$= \frac{3}{2} - \frac{T(c)^2}{2\sigma_w^2} - 2T(c) + \widetilde{\delta}_{h.o.t}(\kappa(x^{c,i}, x^{c,j})) \tag{134}$$

- **Case** $c \neq c'$:

$$\kappa(x^{c,i}, x^{c',j}) = \sqrt{\frac{1}{\sigma_w^2 (x^{c,i})^2 (x^{c',j})^2} + \frac{2}{(x^{c',j})^2} + \frac{2}{(x^{c,i})^2}} \tag{135}$$

$$= \frac{1}{2} + \frac{1}{2\sigma_w^2 (x^{c,i})^2 (x^{c',j})^2} + \frac{1}{(x^{c',j})^2} + \frac{1}{(x^{c,i})^2} + \xi''_{h.o.t} \tag{136}$$

Thus, based on Lemma C.4, we get:

$$\mathbb{E}\left[\kappa(x^{c,i}, x^{c',j})\right] = \frac{1}{2} + \mathbb{E}\left[\frac{1}{2\sigma_w^2 (x^{c,i})^2 (x^{c',j})^2}\right] + \mathbb{E}\left[\frac{1}{(x^{c',j})^2}\right] + \mathbb{E}\left[\frac{1}{(x^{c,i})^2}\right] + \mathbb{E}[\xi''_{h.o.t}] \tag{137}$$

$$= \frac{1}{2} + \frac{1}{2\sigma_w^2}\mathbb{E}\left[\frac{1}{(x^{c,i})^2}\right]\mathbb{E}\left[\frac{1}{(x^{c',j})^2}\right] + \mathbb{E}\left[\frac{1}{(x^{c',j})^2}\right] + \mathbb{E}\left[\frac{1}{(x^{c,i})^2}\right] + \mathbb{E}[\xi''_{h.o.t}] \tag{138}$$

$$= \frac{1}{2} + \frac{T(c)T(c')}{2\sigma_w^2} + T(c') + T(c) + \mathbb{E}[\xi''_{h.o.t}]. \tag{139}$$

This gives us:

$$\mathbb{E}\left[\frac{1}{\kappa(x^{c,i}, x^{c',j})}\right] = \mathbb{E}\left[\frac{1}{1 + \left(-\frac{1}{2} + \frac{T(c)T(c')}{2\sigma_w^2} + T(c') + T(c) + \mathbb{E}[\xi''_{h.o.t}]\right)}\right] \tag{140}$$

$$= 1 - \left(-\frac{1}{2} + \frac{T(c)T(c')}{2\sigma_w^2} + T(c') + T(c) + \mathbb{E}[\xi''_{h.o.t}]\right) + \delta'_{h.o.t}(\kappa(x^{c,i}, x^{c,j})) \tag{141}$$

$$= \frac{3}{2} - \frac{T(c)T(c')}{2\sigma_w^2} - T(c) - T(c') + \widetilde{\delta}_{h.o.t}(\kappa(x^{c,i}, x^{c',j})) \tag{142}$$

Finally, the cases for the expected value of the kernel can be given as:

$$\mathbb{E}\left[\Theta_{NTK-Erf}^{(2)}(x^{c,i}, x^{c',j})\right] = \begin{cases} \mathbb{E}\left[\sigma_w^2 Q_{GP-Erf}^{(1)}(x^{c,i}, x^{c,j})\right] + \mathbb{E}\left[\frac{4\sigma_w}{\pi\kappa(x^{c,i}, x^{c,j})}\right] & c = c' \\ \mathbb{E}\left[\sigma_w^2 Q_{GP-Erf}^{(1)}(x^{c,i}, x^{c',j})\right] - \mathbb{E}\left[\frac{4\sigma_w}{\pi\kappa(x^{c,i}, x^{c',j})}\right] & c \neq c' \end{cases}, \tag{143}$$

From (87), we know that:

$$\mathbb{E}\left[Q_{GP-Erf}^{(1)}(x^{c,i}, x^{c',j})\right]$$

$$\approx \begin{cases} 1 - \frac{T(c)}{2\sigma_w^2} + \Delta_{h.o.t}^{(1)}(c) & \text{if } c = c', i = j \\ 1 - \frac{T(c)}{2\sigma_w^2} - \frac{T(c)^2}{16\sigma_w^4} + \Delta_{h.o.t}^{(2)}(c) & \text{if } c = c', i \neq j \\ 1 - \frac{T(c)+T(c')}{4\sigma_w^2} - \frac{T(c)T(c')}{16\sigma_w^4} + \Delta_{h.o.t}^{(3)}(c, c') & \text{if } c \neq c' \end{cases} \tag{144}$$

To simplify the presentation, we can ignore the higher-order terms and obtain:

$$\mathbb{E}\left[\Theta_{NTK-Erf}^{(2)}(x^{c,i}, x^{c',j})\right] \tag{145}$$

$$\approx \begin{cases} \sigma_w^2\left(1 - \frac{T(c)}{2\sigma_w^2}\right) + \frac{4\sigma_w}{\pi}\left(\frac{3}{2} - 2T(c)\right) & c = c'; i = j \\ \sigma_w^2\left(1 - \frac{T(c)}{2\sigma_w^2} - \frac{T(c)^2}{16\sigma_w^4}\right) + \frac{4\sigma_w}{\pi}\left(\frac{3}{2} - \frac{T(c)^2}{2\sigma_w^2} - 2T(c)\right), & c = c', i \neq j \\ \sigma_w^2\left(1 - \frac{T(c)+T(c')}{4\sigma_w^2} - \frac{T(c)T(c')}{16\sigma_w^4}\right) - \frac{4\sigma_w}{\pi}\left(\frac{3}{2} - \frac{T(c)T(c')}{2\sigma_w^2} - T(c) - T(c')\right), & c \neq c' \end{cases} \tag{146}$$

Observe that the order of the $T(c)$ terms involved here resemble that of the NNGP scenario in (87). Thus, we can make similar conclusions regarding the role of the order of $\mu_c, \sigma_c$ in determining the value of $\mathbb{E}\left[\mathcal{NC}_1(\mathbf{H})\right]$.

## F  Activation Variability Relative to Data

In this section, we introduce a relative measure of activation variability collapse with respect to the data. First, we begin by defining the within-class and between-class data covariance matrices $\mathbf{\Sigma}_W(\mathbf{X}), \mathbf{\Sigma}_B(\mathbf{X}) \in \mathbb{R}^{d_0 \times d_0}$ for the data samples as:

$$\mathbf{\Sigma}_W(\mathbf{X}) = \frac{1}{N}\sum_{c=1}^{C}\sum_{i=1}^{n_c}\left(\mathbf{x}^{c,i} - \overline{\mathbf{x}}^c\right)\left(\mathbf{x}^{c,i} - \overline{\mathbf{x}}^c\right)^\top; \quad \mathbf{\Sigma}_B(\mathbf{X}) = \frac{1}{C}\sum_{c=1}^{C}\left(\overline{\mathbf{x}}^c - \overline{\mathbf{x}}^G\right)\left(\overline{\mathbf{x}}^c - \overline{\mathbf{x}}^G\right)^\top, \tag{147}$$

where $\overline{\mathbf{x}}^c = \frac{1}{n_c}\sum_{i=1}^{n_c}\mathbf{x}^{c,i}, \forall c \in [C]$ and $\overline{\mathbf{x}}^G = \frac{1}{N}\sum_{c=1}^{C}\sum_{i=1}^{n_c}\mathbf{x}^{c,i}$ represent the data class mean vectors and the data global mean vector respectively.

**Definition F.1.** *Set a small $\tau > 0$. The variability collapse relative to the data is given by:*

$$\mathcal{NC}_1(\mathbf{H}|\mathbf{X}) := \frac{\mathcal{NC}_1(\mathbf{H})}{\mathcal{NC}_1(\mathbf{X}) + \tau}, \quad \text{where } \mathcal{NC}_1(\mathbf{X}) := \frac{\text{tr}(\boldsymbol{\Sigma}_W(\mathbf{X}))}{\text{tr}(\boldsymbol{\Sigma}_B(\mathbf{X}))} \tag{148}$$

The constant $\tau$ prevents numerical instabilities. Through this approach, we capture the extent of variability collapse of activation features relative to the variability collapse of the data samples itself.

**Corollary F.2.** *Under Assumptions 1-3 (as per Section 5.3), let $\phi(\cdot)$ be the ReLU activation, and the limiting NNGP kernel be $Q_{GP-ReLU}^{(1)}(\mathbf{x}^{c,i}, \mathbf{x}^{c',j}) = \mathbf{h}^{c,i\top}\mathbf{h}^{c',j}$, then:*

$$\frac{\mathbb{E}\left[\mathcal{NC}_1(\mathbf{H})\right]}{\mathbb{E}\left[\mathcal{NC}_1(\mathbf{X})\right]} \approx 1 - \frac{\frac{2}{N^2}\prod_{c=1}^{2}n_c\mu_c}{\left(\sum_{c=1}^{2}\frac{\mu_c^2}{2} - \frac{n_c^2\mu_c^2}{N^2}\right)} \tag{149}$$

*Proof.* To keep the derivation similar to those for the kernel formulation in equation 45, we consider a simplified kernel on $\mathbf{X}$ (identity feature map):

$$K_{data}(x^{c,i}, x^{c',j}) = x^{c,i}x^{c',j}. \tag{150}$$

Additionally, since $\mathbf{x}^{c,i}$ are 1-d random variables, the expected value of the kernel is given by:

$$\mathbb{E}\left[K_{data}(x^{c,i}, x^{c',j})\right] = \begin{cases} \sigma_c^2 + \mu_c^2 & \text{if } c = c', i = j \\ \mu_c^2 & \text{if } c = c', i \neq j \\ \mu_c\mu_{c'} & \text{if } c \neq c' \end{cases} \tag{151}$$

We use Lemma C.3 with cases $V^{(1)}(c) = \sigma_c^2 + \mu_c^2$, $V^{(2)}(c) = \mu_c^2$ and $V^{(3)}(c, c') = \mu_c\mu_{c'}$ to obtain:

$$\mathbb{E}\left[\mathcal{NC}_1(\mathbf{X})\right] = \frac{\mathbb{E}\left[\text{tr}(\Sigma_W(\mathbf{X}))\right]}{\mathbb{E}\left[\text{tr}(\Sigma_B(\mathbf{X}))\right]} = \frac{\sum_{c=1}^{2}\frac{n_c\mu_c^2 + n_c\sigma_c^2}{N} - \frac{n_c^2\mu_c^2 + n_c\sigma_c^2}{2n_c^2}}{\left(\sum_{c=1}^{2}\frac{n_c^2\mu_c^2 + n_c\sigma_c^2}{2n_c^2} - \frac{n_c^2\mu_c^2 + n_c\sigma_c^2}{N^2}\right) - \frac{2}{N^2}\prod_{c=1}^{2}n_c\mu_c} + \Delta_{h.o.t}^X \tag{152}$$

Finally, the ratio $\frac{\mathbb{E}\left[\mathcal{NC}_1(\mathbf{H})\right]}{\mathbb{E}\left[\mathcal{NC}_1(\mathbf{X})\right]}$ for `ReLU` (Theorem 5.1) with large enough $n_c \gg 1$ is given by:

$$\frac{\mathbb{E}\left[\mathcal{NC}_1(\mathbf{H})\right]}{\mathbb{E}\left[\mathcal{NC}_1(\mathbf{X})\right]} = \frac{\sum_{c=1}^{2}\frac{n_c\mu_c^2 + n_c\sigma_c^2}{N} - \frac{\mu_c^2}{2}}{\left(\sum_{c=1}^{2}\frac{\mu_c^2}{2} - \frac{n_c^2\mu_c^2}{N^2}\right)} \cdot \frac{\left(\sum_{c=1}^{2}\frac{\mu_c^2}{2} - \frac{n_c^2\mu_c^2}{N^2}\right) - \frac{2}{N^2}\prod_{c=1}^{2}n_c\mu_c}{\sum_{c=1}^{2}\frac{n_c\mu_c^2 + n_c\sigma_c^2}{N} - \frac{\mu_c^2}{2}} + \Delta_{h.o.t}'. \tag{153}$$

$$= \frac{\left(\sum_{c=1}^{2}\frac{\mu_c^2}{2} - \frac{n_c^2\mu_c^2}{N^2}\right) - \frac{2}{N^2}\prod_{c=1}^{2}n_c\mu_c}{\left(\sum_{c=1}^{2}\frac{\mu_c^2}{2} - \frac{n_c^2\mu_c^2}{N^2}\right)} + \Delta_{h.o.t}' \tag{154}$$

$$= 1 - \frac{\frac{2}{N^2}\prod_{c=1}^{2}n_c\mu_c}{\left(\sum_{c=1}^{2}\frac{\mu_c^2}{2} - \frac{n_c^2\mu_c^2}{N^2}\right)} + \Delta_{h.o.t}' \tag{155}$$

$\square$

To better understand the result, let us consider the balanced class scenario where $n_1 = n_2 = n = N/2$. This results in a ratio of $\approx 1 - (2\mu_1\mu_2)/(\mu_1^2 + \mu_2^2)$. Furthermore, if $|\mu_1| = |\mu_2|$ (so $\mu_1 = -\mu_2$), then the ratio $\approx 2$.

Thus, it emphasizes the interplay between class imbalance/balance and the values of expected class means on the relative variability collapse.

• **Addressing misleading $\mathcal{NC}_1(\mathbf{H})$ values:** Consider the case where $\sigma_1, \sigma_2 \to 0$. Then Theorem 5.1 for $Q_{GP-ReLU}$ indicates that $\mathbb{E}\left[\mathcal{NC}_1(\mathbf{H})\right] \to 0$ (considering smaller fluctuations from $\Delta_{h.o.t}$) in the balanced class setting. Such an observation can be misleading if one were to ignore $\mathcal{NC}_1(\mathbf{X})$. For instance, such an empirical result while training deep neural networks fails to differentiate between settings where the network learned meaningful features and learned to classify complex datasets or was simply able to leverage the already collapsed data vectors. This applies to `Erf` activation as well. We justify this argument with the following experiment. For a sample size $N$ chosen from $\{128, 256, 512, 1024\}$, and input dimension $d_0$ chosen from $\{1, 2, 8, 32, 128\}$, we sample the vectors $\mathbf{x}^{1,i} \sim \mathcal{N}(-10 * \mathbf{1}_{d_0}, \mathbf{I}_{d_0}), i \in [N/2]$ for class 1 and $\mathbf{x}^{2,j} \sim \mathcal{N}(10 * \mathbf{1}_{d_0}, \mathbf{I}_{d_0}), j \in [N/2]$ for class 2 as our dataset. From Figure 6a, 6b, observe that $\mathcal{NC}_1(\mathbf{H}|\mathbf{X})$ values for $Q_{GP-Erf}$ can be orders of magnitude larger than $\mathcal{NC}_1(\mathbf{H})$, and for high-dimensions $\mathcal{NC}_1(\mathbf{H}|\mathbf{X}) > 1$. Essentially, the raw data is 'more' collapsed than the activations in these settings. Similar observations can be made for the NTK $\Theta_{NTK-Erf}$ in Figure 6c, 6d.

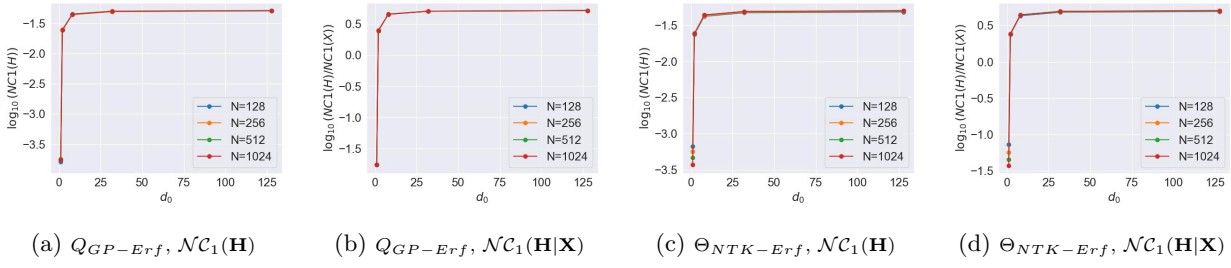

(a) $Q_{GP-Erf}, \mathcal{NC}_1(\mathbf{H})$    (b) $Q_{GP-Erf}, \mathcal{NC}_1(\mathbf{H}|\mathbf{X})$    (c) $\Theta_{NTK-Erf}, \mathcal{NC}_1(\mathbf{H})$    (d) $\Theta_{NTK-Erf}, \mathcal{NC}_1(\mathbf{H}|\mathbf{X})$

Figure 6: $\mathcal{NC}_1(\mathbf{H}), \mathcal{NC}_1(\mathbf{H}|\mathbf{X})$ of $Q_{GP-Erf}^{(1)}$ and $\Theta_{NTK-Erf}^{(2)}$. The dimension $d_0$ on the x-axis is chosen from $\{1, 2, 8, 32, 128\}$. For a particular $N$, we sample the vectors $\mathbf{x}^{1,i} \sim \mathcal{N}(-10 * \mathbf{1}_{d_0}, \mathbf{I}_{d_0}), y^{1,i} = -1, i \in [N/2]$ for class 1 and $\mathbf{x}^{2,j} \sim \mathcal{N}(10 * \mathbf{1}_{d_0}, \mathbf{I}_{d_0}), y^{2,j} = 1, j \in [N/2]$ for class 2.

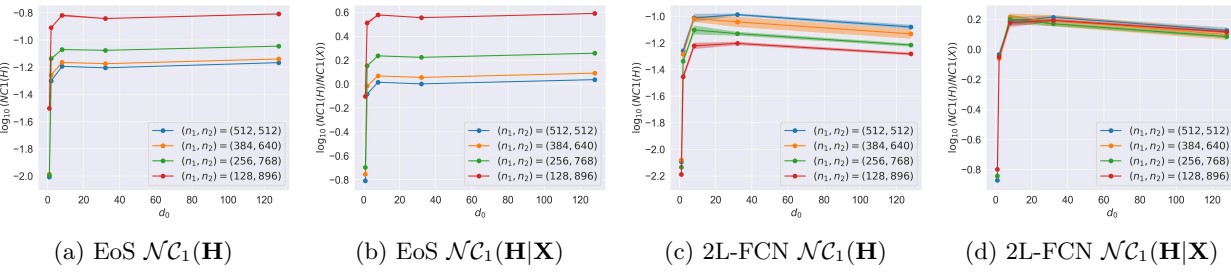

(a) EoS $\mathcal{NC}_1(\mathbf{H})$    (b) EoS $\mathcal{NC}_1(\mathbf{H}|\mathbf{X})$    (c) 2L-FCN $\mathcal{NC}_1(\mathbf{H})$    (d) 2L-FCN $\mathcal{NC}_1(\mathbf{H}|\mathbf{X})$

Figure 7: $\mathcal{NC}_1(\mathbf{H}), \mathcal{NC}_1(\mathbf{H}|\mathbf{X})$ of the adaptive kernel (EoS) with final annealing factor $d_1 = 500$ and 2L-FCN with $d_1 = 500$ and `Erf` activation. The dimension $d_0$ on the x-axis is chosen from $\{1, 2, 8, 32, 128\}$. For a tuple $(n_1, n_2)$ such that $n_1 + n_2 = N = 1024$, we sample the vectors $\mathbf{x}^{1,i} \sim \mathcal{N}(-2 * \mathbf{1}_{d_0}, 0.25 * \mathbf{I}_{d_0}), y^{1,i} = -1, i \in [n_1]$ for class 1 and $\mathbf{x}^{2,j} \sim \mathcal{N}(2 * \mathbf{1}_{d_0}, 0.25 * \mathbf{I}_{d_0}), y^{2,j} = 1, j \in [n_2]$ for class 2.

## G    Training Details and Additional Experiments

**Setup:** We train a 2L-FCN with $d_1 = 500, \sigma_w = 1, \sigma_b = 0$ and `Erf` activation using (vanilla) Gradient Descent with a learning rate of $10^{-3}$ and weight-decay $10^{-6}$ for 1000 steps to reach the terminal phase of training (see Figure 8) for plots with $N = 1024$ and vectors $\mathbf{x}^{1,i} \sim \mathcal{N}(-2 * \mathbf{1}_{d_0}, 0.25 * \mathbf{I}_{d_0}), y^{1,i} = -1, i \in [N/2]$ for class 1 and $\mathbf{x}^{2,j} \sim \mathcal{N}(2 * \mathbf{1}_{d_0}, 0.25 * \mathbf{I}_{d_0}), y^{2,j} = 1, j \in [N/2]$ for class 2). The `ReLU` activation experiments use a learning rate of $10^{-4}$. To numerically solve the EoS, we employ the approach described in Section 6.2. All the experiments in this paper were executed on a machine with 16 GB of host memory and 8 CPU cores. Experiments with the EoS on datasets of varying dimensions and sample sizes took the longest time $\approx 1$ hour.

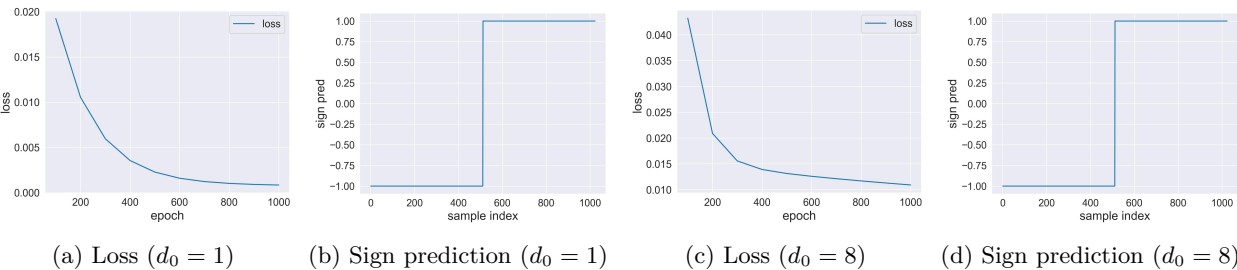

(a) Loss ($d_0 = 1$)    (b) Sign prediction ($d_0 = 1$)    (c) Loss ($d_0 = 8$)    (d) Sign prediction ($d_0 = 8$)

Figure 8: Training loss and predictions (sign) of the 2L-FCN model with $d_1 = 500$ and Erf activation.

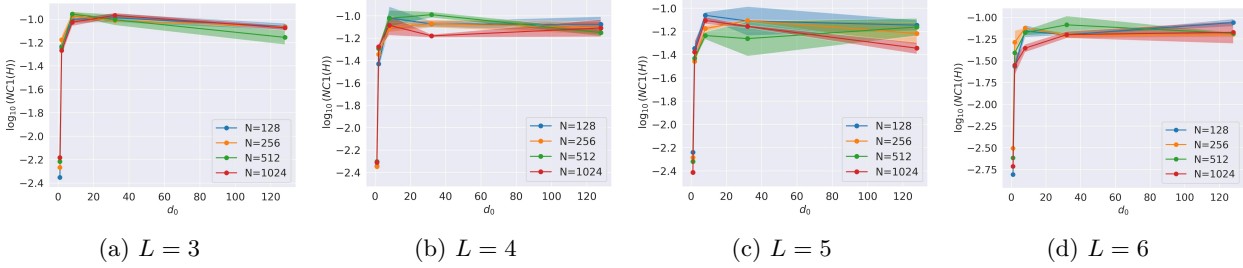

(a) $L = 3$    (b) $L = 4$    (c) $L = 5$    (d) $L = 6$

Figure 9: $\mathcal{NC}_1(\mathbf{H})$ of deeper FCN networks with `Erf` activation and hidden later width 500. The dimension $d_0$ on the x-axis is chosen from $\{1, 2, 8, 32, 128\}$. For a particular $N$, we sample the vectors $\mathbf{x}^{1,i} \sim \mathcal{N}(-2 * \mathbf{1}_{d_0}, 0.25 * \mathbf{I}_{d_0}), y^{1,i} = -1, i \in [N/2]$ for class 1 and $\mathbf{x}^{2,j} \sim \mathcal{N}(2 * \mathbf{1}_{d_0}, 0.25 * \mathbf{I}_{d_0}), y^{2,j} = 1, j \in [N/2]$ for class 2.

**On increasing the number of classes with separable data.** Similar to the formulation of $\mathcal{D}_1(N, d_0)$ for $C = 2$, we formulate $\mathcal{D}_2(N, d_0)$ for $C = 4$ as follows:

$$
\begin{aligned}
\mathcal{D}_2(N, d_0) = \ & \left\{ (\mathbf{x}^{1,i} \sim \mathcal{N}(-6 * \mathbf{1}_{d_0}, 0.25 * \mathbf{I}_{d_0}), y^{1,i} = -3), \forall i \in [N/4]) \right\} \\
& \cup \left\{ (\mathbf{x}^{2,j} \sim \mathcal{N}(-2 * \mathbf{1}_{d_0}, 0.25 * \mathbf{I}_{d_0}), y^{2,j} = -1), \forall j \in [N/4]) \right\} \\
& \cup \left\{ (\mathbf{x}^{3,k} \sim \mathcal{N}(2 * \mathbf{1}_{d_0}, 0.25 * \mathbf{I}_{d_0}), y^{3,k} = 1), \forall k \in [N/4]) \right\} \\
& \cup \left\{ (\mathbf{x}^{4,l} \sim \mathcal{N}(6 * \mathbf{1}_{d_0}, 0.25 * \mathbf{I}_{d_0}), y^{4,l} = 3), \forall l \in [N/4]) \right\}.
\end{aligned}
\tag{156}
$$

The trends in $\mathcal{NC}_1(\mathbf{H})$ for EoS and 2L-FCN hold even for datasets $\mathcal{D}_2(N, d_0)$ with $C > 2$. Observe from Figure 10 that the $\mathcal{NC}_1(\mathbf{H})$ values for the kernels $Q_{GP-Erf}, \Theta_{NTK-Erf}$, EoS and for 2L-FCN are consistently lower than the $C = 2$ case as shown in Figures 2a, 2b, 2c, 4b, even when comparing them at $d_0 = 1$. We believe this is because $\mathcal{D}_2(N, d_0)$ is constructed by adding 2 new classes to $\mathcal{D}_1(N, d_0)$ whose distance between the means is much larger than the within class co-variances (see Figure 11). Nonetheless, similar to the $C = 2$ case, the EoS deviates from the NNGP behavior and tends to approximate the trends of 2L-FCN for high dimensional data with a sufficiently large sample size.

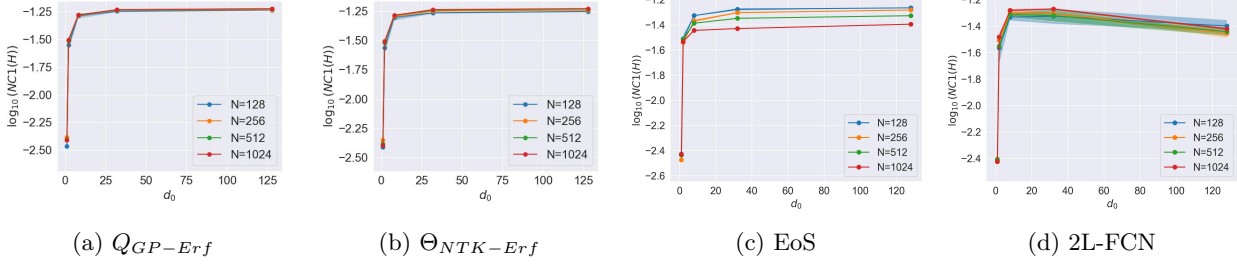

(a) $Q_{GP-Erf}$    (b) $\Theta_{NTK-Erf}$    (c) EoS    (d) 2L-FCN

Figure 10: $\mathcal{NC}_1(\mathbf{H})$ of the limiting kernels, adaptive kernel (EoS) with final annealing factor 500 and 2L-FCN with $d_1 = 500$ and Erf activation on dataset $\mathcal{D}_2(N, d_0)$ (as per (156)).

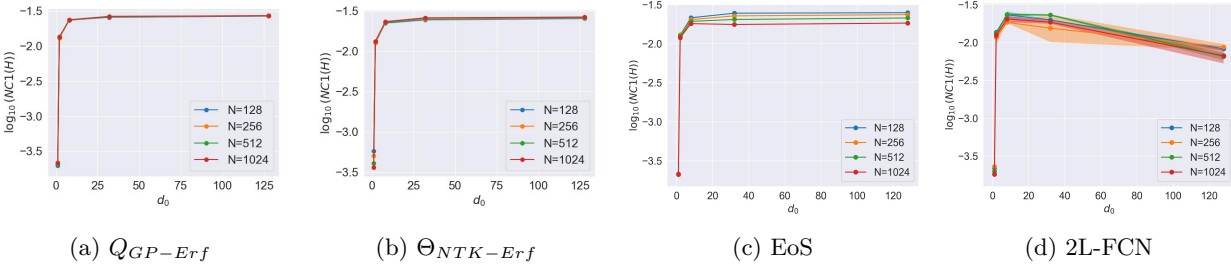

(a) $Q_{GP-Erf}$     (b) $\Theta_{NTK-Erf}$     (c) EoS     (d) 2L-FCN

Figure 11: $\mathcal{NC}_1(\mathbf{H})$ of the limiting kernels, adaptive kernel (EoS) with final annealing factor $d_1 = 500$ and 2L-FCN with $d_1 = 500$ and `Erf` activation. The dimension $d_0$ on the x-axis is chosen from $\{1, 2, 8, 32, 128\}$. For a particular $N$, we sample the vectors $\mathbf{x}^{1,i} \sim \mathcal{N}(-6 * \mathbf{1}_{d_0}, 0.25 * \mathbf{I}_{d_0}), y^{1,i} = -1, i \in [N/2]$ for class 1, and $\mathbf{x}^{4,j} \sim \mathcal{N}(6 * \mathbf{1}_{d_0}, 0.25 * \mathbf{I}_{d_0}), y^{2,j} = 1, j \in [N/2]$ for class 2.

**Class imbalance hurts EoS in approximating 2L-FCN.** One of the key conditions for the EoS to approximate the $\mathcal{NC}_1(\mathbf{H})$ behavior of a 2L-FCN is the presence of sufficient data samples (Seroussi et al., 2023) (depending on the scale of the input dimension $d_0$ as shown above). Thus, extreme class imbalances can lead to biased $\mathcal{NC}_1(\mathbf{H})$ trends in EoS. To this end, we consider a collection of imbalanced datasets based on $\mathcal{D}_1(N, d_0)$ where $N = 2048$ is split into two classes as follows: **Case 1:** $(768, 1280)$, **Case 2:** $(512, 1536)$, **Case 3:** $(256, 1792)$. We observe that as the imbalance ratio increases, the magnitude of $\mathcal{NC}_1(\mathbf{H})$ in 2L-FCN and kernel methods show opposing trends (see Figure 12). We make a similar observation by considering $\mathcal{D}_2(N, d_0)$ with $C = 4$ and $N = 1024$ with 3 different imbalance ratios as in Figure 13.

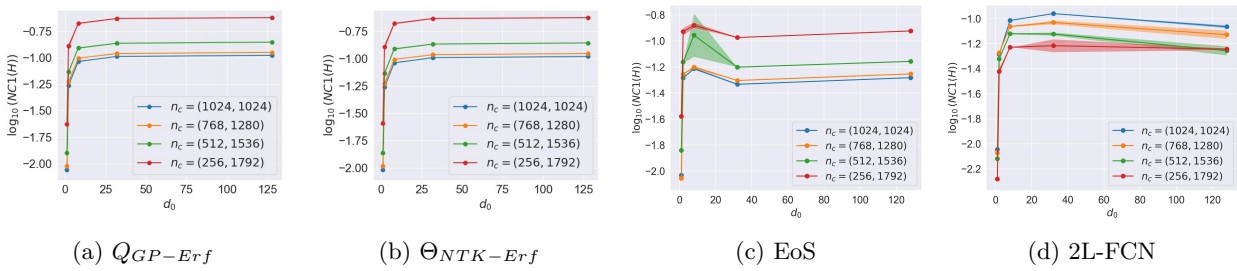

(a) $Q_{GP-Erf}$     (b) $\Theta_{NTK-Erf}$     (c) EoS     (d) 2L-FCN

Figure 12: $\mathcal{NC}_1(\mathbf{H})$ of the limiting kernels, adaptive kernel (EoS) with final annealing factor $d_1 = 500$ and 2L-FCN with $d_1 = 500$ and `Erf` activation. The dimension $d_0$ on the x-axis is chosen from $\{1, 2, 8, 32, 128\}$. For a tuple $n_c = (n_1, n_2)$ such that $n_1 + n_2 = N = 2048$, we sample the vectors $\mathbf{x}^{1,i} \sim \mathcal{N}(-2 * \mathbf{1}_{d_0}, 0.25 * \mathbf{I}_{d_0}), y^{1,i} = -1, i \in [n_1]$ for class 1 and $\mathbf{x}^{2,j} \sim \mathcal{N}(2 * \mathbf{1}_{d_0}, 0.25 * \mathbf{I}_{d_0}), y^{2,j} = 1, j \in [n_2]$ for class 2.

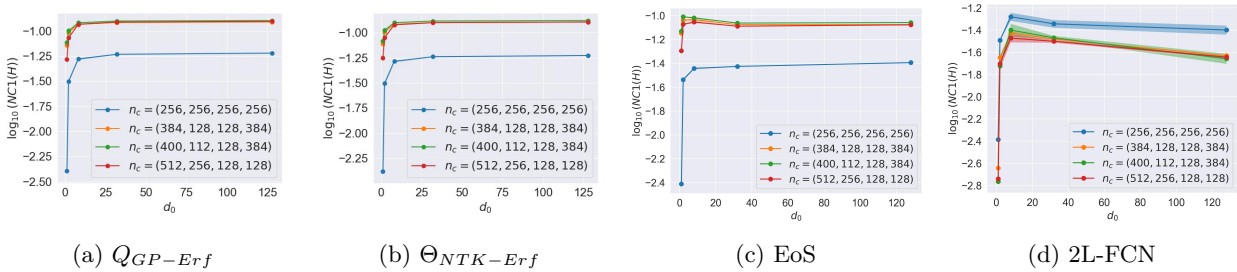

(a) $Q_{GP-Erf}$     (b) $\Theta_{NTK-Erf}$     (c) EoS     (d) 2L-FCN

Figure 13: $\mathcal{NC}_1(\mathbf{H})$ of the limiting kernels, adaptive kernel (EoS) with final annealing factor $d_1 = 500$ and 2L-FCN with $d_1 = 500$ and `Erf` activation. The dimension $d_0$ on the x-axis is chosen from $\{1, 2, 8, 32, 128\}$. For a tuple $n_c = (n_1, n_2, n_3, n_4)$ such that $n_1 + n_2 + n_3 + n_4 = N = 1024$, we sample the vectors $\mathbf{x}^{1,i} \sim \mathcal{N}(-6 * \mathbf{1}_{d_0}, 0.25 * \mathbf{I}_{d_0}), y^{1,i} = -3, i \in [n_1]$, $\mathbf{x}^{2,j} \sim \mathcal{N}(-2 * \mathbf{1}_{d_0}, 0.25 * \mathbf{I}_{d_0}), y^{2,j} = -1 j \in [n_2]$, $\mathbf{x}^{3,k} \sim \mathcal{N}(2 * \mathbf{1}_{d_0}, 0.25 * \mathbf{I}_{d_0}), y^{3,k} = 1, k \in [n_3]$ and $\mathbf{x}^{4,l} \sim \mathcal{N}(6 * \mathbf{1}_{d_0}, 0.25 * \mathbf{I}_{d_0}), y^{4,l} = 3, l \in [n_4]$.

