# OpenReview forum: "Can Kernel Methods Explain How the Data Affects Neural Collapse?"
_TMLR — Accepted by TMLR_

### Review · Reviewer_NTBv · 2025-03-12

**Summary Of Contributions:**

This paper studies the neural collapse for kernel methods. The authors first derive the NC1 matric for general kernel methods, applying the results to NTK and NNGP kernel, and then doing experiments to support the theoretical findings. Then the authors discuss the data-dependent kernels and also conduct experiments on data-dependent kernels.

**Audience:**

Yes

**Claims And Evidence:**

Yes

**Requested Changes:**

Could you address the **Weaknesses** part? I don't have further request changes.

**Strengths And Weaknesses:**

**Strengths**
1. This paper studies the effect of data distribution on NC, which I think it's interesting
2. I think the study of NC of kernel learning methods, especially with data-dependent kernels is an interesting topic
3.  In section 6, the authors empirically find that by adding data-dependent correction terms to the NNGP kernels improve the NC1 metric, which is an interesting finding in my opinion.
4. I think this paper is well-written and easy to follow

**Weaknesses**

1. My major concern is that I don't see the point of directly considering the NTK or NNGP kernels as the feature kernel, and studying the NC1 of NTK or NNGP kernels. If one consider the feature kernel is the  NNGP or NTK, it implies that the features **are fixed** at initialization, meaning that the features are just "random features" and there's no feature learning in this case. It is kind of obvious that NC1 will not occur in these cases. Concretely, in the NNGP case for example,  the feature $h(x)$ of a data point $x$ is just $h(x) = \phi( W^{(1)} x),$ where $W^{(1)}$ is the i.i.d Gaussian matrix at initialization. The NTK case is similar with a more complicated expression of the feature.

2. I don't see the utility of Theorem 5.1 in general.  First of all, the assumptions that there's only two classes of $1$-dim Gaussian data, and $|\mu| \gg \sigma$ is kind of restrictive, since this means the dataset is already well-separated. In particular, the expected NC1 is of the same order of the variance as discussed in the part below Theorem 5.1, meaning that the  kernels does not improve the NC1 of the original dataset. Besides, under this setting, the main information from Theorem that NC1 for the NTK and NNGP kernel is the same up to higher-order terms are again not surprising in my opinion, since in both NTK and NNGP case, the feature is again a "random feature", and it is not hard to imagine that the NC1 of feature is equivalent to the NC1 of the datasets.

3. Another main issue of this paper is that the authors do not consider any training on the features (since the feature kernel is the NNGP kernel or the NTK kernel). However, as mentioned in the original NC paper [1], NC is considered to be a phenomenon occuring at the ending phase of the training. Thus, I think the usefulness of this paper's results in understanding NC might be limited.

---

> ### Author Response · Authors · 2025-03-26
>
> We sincerely appreciate the reviewer's feedback and suggestions.
>
> **1. Regarding the role of NNGP/NTK for NC analysis:** Recent work, such as [1], has attempted to connect NTK and NC (based on the NTK alignment literature), so we believe that it is essential to rigorously characterize the shortcomings of the limiting NTK. Furthermore, our thought process/motivation behind analyzing these kernels is as follows.
>
> - We do not aim to explain NC via limiting NNGP/NTK but instead use them as baselines to motivate our analysis on EoS and also follow a theme of comparatively studying kernels corresponding to NNs (i.e $NNGP \to NTK \to EoS \to 2L-FCN$)
>
> - It is known that limiting NNGP/NTK pertains to infinitely wide NNs at initialization and training with very small learning rates respectively. To this end, (1) we consider NNGP to study the NC properties at NN initialization, (2) and since the NTK feature map takes into consideration the architecture and the fact that GD optimization is used (although not data-aware), it potentially allows us to gain insights beyond NNGP (as done in different fields). From this point of view, our results show no reduction of NC1 *when comparing* NTK to NNGP. Thus rigorously justifying the move to data-aware kernels.
>
> **2. Regarding the Theorem 5.1** We want to underscore that Theorem 5.1 allows us to *mathematically prove* the limitation of NTK in the context of NC analysis. As discussed in the above point, since recent efforts have started analyzing the NTK in the context of NC, this theorem allows us to characterize its limitations.
>
> Furthermore, although Theorem 5.1 relies on well-separated classes and $d_0=1$, it rigorously establishes the role of ReLU activation in computing the NC1 value and also facilitates similar calculations for the Erf activation (Appendix E). Formally, the main takeaways from the theorem correspond to the similar $NC1(H)$ of NTK/NNGP and the dependency of $NC1(H)$ on $\mu_c, \sigma_c$ of the classes and its interplay with the choice of activation.
>
> Additionally, this theorem allows us to formally show in Corollary F.2 (Appendix F) that when considering the NC1 metric of features relative to the NC1 of data, the former can be higher than the latter.  See Section 7 and also Appendix F for more discussion on this aspect.
>
> **3. Regarding trained features and kernels:** We would like to first emphasize that :
>
> (1) our paper only aims to understand if and when kernel methods can be suitable for analyzing the NC properties that actual NNs (such as the 2L-FCN) exhibit during training and
>
> (2) The equations of state (EoS) are indeed updated using the annealing procedure (Section 6.2) to model the NN training/feature update process.
>
> Since NC occurs towards the end of training when zero training error is reached, we indeed tracked the 2L-FCN's training loss under different settings in the paper and verified that the classifier reached zero training error and the training loss was also low. Current approaches to model such behavior rely on the UFM approach which completely ignores the data being used for training. This is a major problem since it fails to explain the different $NC1(H)$ values that various NNs exhibit when trained on datasets of varying complexity (see Papyan et al.).
>
> Currently, there are no other richer models to study NC other than UFM and this paper aims to explore the possibility of employing kernels (data dependent/independent) for such a study. This is especially required since previous works such as [1] have already started exploring the empirical NTK as a potential way to explain NC but fail to explain how the class-wise block structure emerges during the early phase of training.
>
> Overall, we believe that the following results from our paper can be useful contributions to the community:
>
> 1. The limiting NTK does not represent more collapsed features than the NNGP and might not be suitable for studying NC. Our key results complement the work of [1] and showcase the limitations of such data-independent kernels.
>
> 2. The role of activation functions has not been extensively studied in the NC literature. This paper theoretically and empirically establishes its role on NC, which was previously not studied with UFMs or even "extended UFMs".
>
> 3. The potential of data-dependent kernels in studying the feature learning aspects of NNs provides a novel perspective to study NC beyond the UFM-based approaches. Although we discuss the limitations in Section 7, future work in this direction can provide significant breakthroughs in theoretically modeling the NN training dynamics.
>
> We believe that these results will open up various future research directions, both in terms of theoretical modeling of the data-dependent kernels as well as new and efficient techniques to solve the system of equations.
>
> [1] Seleznova, Mariia, et al. "Neural (tangent kernel) collapse." Advances in Neural Information Processing Systems 36 (2023): 16240-16270.

---

### Review · Reviewer_DFdc · 2025-03-14

**Summary Of Contributions:**

This work aims at theoretically studying neural collapse for neural networks. While previous theoretical works mostly use the Unconstrained Feature Model (UFM), this work studies it through kernel methods, being notably relevant in the NTK regime when training neural networks.

For Gaussian Process and NTK kernels, it then proposes an expression for neural collapse (being the ratio between inner class variability and interclass variability), and also proposes an analysis through Equation of State (EoS) to get a data-aware kernel.

**Audience:**

Yes

**Claims And Evidence:**

Yes

**Requested Changes:**

I would like the paper to be largely improved in clarity, especially by adding many high level paragraphs explaining the implications of the different theoretical results (see above)

**Strengths And Weaknesses:**

Studying neural collapse with data dependent analyses indeed seems a necessary step to get a better understanding of neural collapse, in which situations does it happen and how it does so.

The paper strongly lacks in clarity in my opinion: in particular, I have difficulties navigating through the different results and interpreting them. In particular, the original intent of the authors is to describe neural collapse in a data-aware fashion. However, the 8 first pages of the paper are dedicated to NNGP and NTK kernels analysis, which are not "data-aware". More importantly, such analyses seem to describe the Neural Collapse at initialization of the training, while neural collapse seems to clearly be a phenomenon appearing during the training of networks (being there at convergence, but not at initialization). Therefore, I do not really understand how all the results before Section 6 serve the original author's intent.

Additionally, I feel that the main sections are a sequence of technical results, but they lack, in my opinion, some higher level interpretation by the authors. More detailed, high level, interpretations of these results would help the reader in understanding this work, and also understanding where we are going while reading the paper.
In the same spirit, I would like an extended introduction to the EoS framework, as it is not yet very common in the literature. In particular, I did not understand whether the EoS equations are supposed to describe the state of the network at convergence, initialization or some point of the training dynamics.

Overall, this notion of training dynamics seems missing in the paper, while it should be key in understanding neural collapse. Even the experiments do not reflect any evolution of the neural collapse throughout training (but only at convergence?).

On the technical aspect of the claimed results, all the maths seem sound, despite some notations/assumptions/definitions that need to be clarified (see below). Overall, my opinion is that this paper writing needs to be largely polished to help the reader navigate through the different results and how they relate to the original intent of the authors.

Also, the data-aware kernel part seems to be the most significant one of the paper (at least when reading the abstract), and I thus think it would be nice to enrich this part (notably by giving a more detailed introduction to the EoS framework and give more detailed interpretations of the results derived here).

-------------
# Minor remarks

- Equations (1) and (2): $\psi$ and $\hat{y}$ are exchangeably used for the same thing. I would recommend to stick with a single notation
- beginning of Section 4: that would be helpful to precise the dimension of the matrix $H$ here
- in Section 5, there sometimes are dot appearing above the kernel functions $Q$. What does it mean?
- Assumptions 1 and 2: what does the $\gg$ notation mean? It is not clear how it intervenes in Theorem 5.1
- Theorem 5.1: what is $Q_{GP}$? Is it $Q_{GP-Erf}$?
- Theorem 5.1: "that vanishes as $\{n_c\}$ increases" -> is it for $\min_c n_c$?
- Key takeaway (section 5.3): I do not see where you "have established another result that shows that training in the lazy regime provably deviates from the practical feature learning of NNs" -> this is clearly the kind of thing we need more discussion about
- same about "these observations empirically verify our theoretical results": I think the comparison is only given in the appendix. It would be nice to have some comparison in the main text then

---

> ### Author Response · Authors · 2025-03-30
>
> We thank the reviewer for the feedback.
>
> **Regarding the intent of the paper:** We would like to kindly emphasize the intent of this study. Typically, existing works analyze NC using UFMs, which are not linked to data at all --- unlike even linear models and kernels. Therefore, in this paper we ask the question: "is it possible to go beyond UFMs with kernel-based analysis?"
>
> - Note that the NTK feature map does take into consideration architecture and the fact that GD optimization is used to train the DNN, even though it is not data-aware. Therefore, potentially it allows to gain insights beyond NNGP (as done in different fields).
> Furthermore, researchers have already attempted to connect NTK and NC in [1].
> - As a starting point, our results show no reduction of NC1 *when comparing* NTK (represents the network trained in the ``lazy regime'') to NNGP (represents the network at initialization) and rigorously justify the move to data-aware kernels. We also explore the recent adaptive kernel and identify the potential of adaptivity together with the current limitations. To conclude, we answer the above question: "is it possible to go beyond UFMs with kernel-based analysis?", as follows: *not without adaptation and provided that more advances would be made in this front.*
>
> [1] Seleznova, Mariia, et al. "Neural (tangent kernel) collapse." Advances in Neural Information Processing Systems 36 (2023): 16240-16270.
>
> **Regarding the interpretation of results:** We will emphasize in the revision the interpretation of our results, as discussed in our answer above. We will also further explain the EoS framework and clarify that it aims to model the state of the DNN at convergence and not during training.
>
> **Regarding the notion of training dynamics and convergence:** Indeed, we focus on the level of collapse at convergence, which is under-researched and cannot be captured by UFM based analysis.
> Existing empirical results show that NC metrics reach plateaus at levels that differ between datasets, architectures, etc. (Papyan et al., 2020), (Han et al., 2022), (Tirer et al., 2023).
> Our kernel-based analysis already shows dependency of the NC1 metric on the dimension of the input, which can be interpreted as the input complexity.
> As we suggest in the paper, we believe that future advances in adaptive kernels can provide significant breakthroughs also in modeling the NN training dynamics.
>
> [1] Vardan Papyan, XY Han, and David L Donoho. Prevalence of neural collapse during the terminal phase of
> deep learning training. Proceedings of the National Academy of Sciences, 117(40):24652–24663, 2020.
>
> [2] XY Han, Vardan Papyan, and David L Donoho. Neural collapse under mse loss: Proximity to and dynamics
> on the central path. In International Conference on Learning Representations, 2022.
>
> [3] Tom Tirer, Haoxiang Huang, and Jonathan Niles-Weed. Perturbation analysis of neural collapse. In International Conference on Machine Learning, pp. 34301–34329. PMLR, 2023.
>
> **Q. in Section 5, there sometimes are dot appearing above the kernel functions Q. What does it mean?**
>
> A. The $\dot{Q}$ represents the derivative kernel as mentioned in Eq 15 and the line above it.
>
> **Q. Assumptions 1 and 2: what does the  $\texttt{>>}$ notation mean? It is not clear how it intervenes in Theorem 5.1**
>
> A. It implies that the $|\mu_c| $ is finite yet very large compared to $\sigma_c$, where $c= \{1,2\}$. Essentially the class means are far apart such that the sign of a sample can indicate the class it belongs to.
>
> **Q. Theorem 5.1: what is $Q_{GP}$ ?**
>
> A. Since Theorem 5.1 corresponds to ReLU activation, $Q_{GP}$ pertains to $Q_{GP-ReLU}$. We will clarify this in the revision.
>
> **Q. Theorem 5.1: "that vanishes as $n_c$ increases" -> is it for $min_c n_c$?**
>
> A. The h.o.t terms involve all $n_c$ terms and thus require all of them to be large so that it is negligible. To this end, since we do not consider a balanced case scenario, considering $min_c n_c$ to be large enough is a valid condition for the h.o.t to be small.
>
>
> Overall, we will apply the remaining minor changes suggested by the reviewer (as per minor remarks) and also provide a high-level overview of the EoS and interpretation of the results (as discussed above). We are happy to provide further clarifications if needed. Thanks!

---

### Review · Reviewer_TjHC · 2025-03-19

**Summary Of Contributions:**

This paper explores an alternative to the unconstrained features model (UFM), a widely used theoretical model for studying neural collapse. The main limitation of UFM is that it assumes unconstrained features, ignoring the role of the input data distribution. This work investigates whether kernels associated with neural networks can offer a more realistic alternative. The authors focus on NC1 (the collapse of features within a class) as a first step, leaving the geometric properties described by NC2–NC4 for future work.

They study three types of kernels: 1) NNGP kernel corresponding to infinitely wide neural networks (NNs) at initialization, 2)NTK corresponding to infinitely wide NNs in the lazy training regime, and 3) Data-aware GP kernel, an alternative kernel that reflects the effect of data on a finite-width neural network, capturing aspects of feature learning.

Theoretically, they show that for binary scalar Gaussian data and a shallow 2-layer network, the expected NC1 values for NNGP and NTK are equal—highlighting a gap with practical training where features learned during training exhibit more collapse than those at initialization, further reinforcing NTK's limitations in capturing true training dynamics.

Empirically, they compare NC1 values from the kernels with those from features learned by training a shallow 2-layer network. The results demonstrate the potential and limitations of kernel-based models in explaining neural collapse and offer comparisons across different setups, e.g., different activation functions or data distributions.

**Audience:**

Yes

**Claims And Evidence:**

Yes

**Requested Changes:**

1.  Since NC1 requires reasonable overparameterization and occurs in the interpolation regime, it would be helpful to confirm and report whether, at the end of training for the shallow networks trained by GD, the model has entered the interpolation/low training-loss regime for the different configurations reported in the paper.

2. The points below might need more careful discussion in order to make the message of the experiments clearer:

    (i) *On NTK limitation*: The paper argues (theoretically and empirically) that NTK limitation is due to the fact that it matches NNGP in terms of the NC1 metric. One confusion can arise from comparing figs 2 and 3 (a) vs (c), where even the NC1 of the features learned by gradient descent and at initialization (characterized by NNGP) are very close as well (In fact, in figure 2, all the three subplots (a), (b), and (c) show the same NC1 trend and value).

    (ii) *On the benefit of data-aware GP*: When discussing the EoS results, it's stated that for larger network dimensions, the data-aware kernel matches both NNGP and NTK closely. Does this suggest that this kernel has the same limitation as NTK in explaining neural collapse?

    In the lower-dimensional case, the behavior changes. However, 1) it also doesn’t fully align with the fully connected case. For different sample sizes $N$, the curves show varying trends, unlike the shallow network experiments in fig. 2, and 2) The higher-dimensional case is more relevant to the neural collapse regime since NC typically emerges under overparameterization.

    (iii) The message of fig. 6 is not very clear: When comparing the collapse of input and kernel features, is this another reason for the limitation of NTK in explaining neural collapse? Or is it a limitation of the shallow NN setup? Specifically, does the same observation ($NC(H|X)>1$) hold for the features of trained shallow networks as well?

3. *Typo:* The caption of Figure 4 refers to subfigures (c) and (d), which do not exist.

**Strengths And Weaknesses:**

The paper is well-organized and easy to follow.

It is also well-motivated: The authors seek to address UFM's shortcomings by leveraging insights from kernel literature to assess whether different kernels offer additional insights or demonstrate other limitations. This is a meaningful step toward developing a more accurate theoretical model.

However, some discussions on the empirical results raise confusion, and further clarification would be helpful. I will expand on this below.

---

> ### Author Response · Authors · 2025-03-26
>
> We sincerely thank the reviewer for providing constructive feedback.
>
> **1. Regarding the training error and loss:** We tracked the 2L-FCN’s training loss under different settings in the paper and verified that the classifier reached zero training error. The training loss was also low. For example: for $d_0 = 1$ with Erf activation, the MSE training loss reaches 0.0012 after 1000 epochs. For higher-dimension data such as $d_0 = 8$, we obtained a loss value of 0.018. We will add the plots and confirm the TPT phase in the revised version of the article.
>
> **2 (i) Regarding the NTK limitation:** We intended to show the NC results for 2L-FCN with a large width such as $d_1=2000$ to confirm that as the width of the NN increases, the NNGP/NTK can indeed approximate the NC behavior (in the ``lazy regime''). However, the differences in NC values between 2L-FCN and NNGP/NTK start to arise when going to finite-width regimes with $d_1=500$. Furthermore, the $d_1=2000$ plots also tend to validate the observation that the EoS resembles the NNGP at such large widths (Figure 4a) (since the correlation matrix $C$ does not change as $d_1$ increases to large values. Eq (18)). We will re-organize the figures and clarify these points in the revised version to avoid such confusion.
>
> **2 (ii) Regarding the data-aware GP:** The initial state of the EoS indeed resembles the NNGP kernel (as per the "Relationship with NNGP" paragraph in Section 6.1). However, a key difference between EoS and NNGP/NTK is that the EoS takes into account the finite values of $d_1$ when computing the covariance matrix $C$ (as per eq 18). This means that for large $d_1$, i.e $d_1 \to
> \infty$, it has the same limitations as NNGP/NTK. But when $d_1 >> 1$ (i.e large but finite), the EoS tends to showcase lower NC values compared to NNGP/NTK.
>
> We acknowledge that the EoS does not exactly resemble the NC trends/values of 2L-FCN (Section 6.3), but our work only aims to convey the message that the EoS is relatively better than NNGP/NTK at exhibiting lower NC values and approximating the 2L-FCN behavior. In particular, we can notice from Figures 2d and 3b that the EoS exhibits lower NC values similar to 2L-FCN which the NNGP/NTK fails to exhibit. However, for the non-linearly separable classes (Figure 5), the EoS and 2L-FCN NC behavior is quite different for high dimensions and is reflective of the shortcomings of the current state of adaptive kernels. Accordingly, as we state in the conclusion, the paper's message is that advancements in modeling NNs by adaptive kernels can benefit NC analysis.
>
> **2 (iii) Regarding the message of NC(H|X):** Thanks for raising this point. Figure 6 showcases the $NC(H|X) > 1$ values for NNGP/NTK  and is an artifact of the setup. In Figure 7, we plot the $NC(H), NC(H|X)$ for the 2L-FCN and EoS (balanced as well as imbalanced cases, albeit for a different choice of $\mu_c, \sigma_c$) and observe that $NC(H|X)$ can indeed be $>1$. This is indeed an artifact of the setup where the data is already well separated. Overall, we aim to convey the message that NC of the input data i.e. $NC(X)$ should also be considered in the NC analysis whenever applicable so that one can quantify how collapsed the kernel/FCN features are relative to the data.
>
> **3. Regarding the typo in caption:** Thanks for pointing it out. We will fix it in the revised article.

---

### Decision · Action_Editor_Ldg4 · 2025-04-20

**Recommendation:** Accept as is

**Comment:**

The paper studies the core component of neural collapse, namely the within-class variability collapse. The UFM, which constitutes the established framework to study neural collapse, is data-agnostic and, to address this issue, the paper proposes to focus kernel methods, considering the NNGP, the NTK and a data-aware GP kernel. The results show the limitation of data-agnostic kernels (NNGP, NTK) and that the data-aware GP kernel exhibits superior collapse.

The rebuttal period has helped clarify some misunderstandings and the most pressing issues raised by the reviewers have been resolved. The three reviewers agree on the overall assessment: although the results are not particularly surprising, they are technically correct and prove an important point concerning the importance of data adaptivity for neural collapse. I concur and, considering the TMLR guidelines, I recommend accepting the paper.

**Audience:**

There is agreement among reviewers that the paper can be of interest to the TMLR audience seeking theoretical tools for explaining neural collapse. I agree with this view.

**Claims And Evidence:**

The paper is technically sound. The round of review-rebuttals has improved the presentation and clarified the scope of the work.